# Stepwise Feature Learning in Self-Supervised Learning

## Abstract

Recent advances in self-supervised learning (SSL) have shown remarkable progress in representation learning. However, SSL models often exhibit shortcut learning phenomenon, where they exploit dataset-specific biases rather than learning generalizable features, sometimes leading to severe over-optimization on particular datasets. We present a theoretical framework that analyzes this shortcut learning phenomenon through the lens of *extent bias* and *amplitude bias*. By investigating the relations among extent bias, amplitude bias, and learning priorities in SSL, we demonstrate that learning dynamics is fundamentally governed by the dimensional properties and amplitude of features rather than their semantic importance. Our analysis reveals how the eigenvalues of the feature cross-correlation matrix influence which features are learned earlier, providing insights into why models preferentially learn shortcut features over more generalizable features.

## 1   Introduction

While deep neural networks have shown remarkable success in various learning tasks, recent studies have revealed a concerning trend: models often exploit unexpected learning behavior, particularly shortcut learning, which tends to take easier but potentially less reliable paths to solve general tasks [13]. For example, in image classification tasks, models tend to learn earlier larger background features than smaller foreground objects [17], potentially leading them to classify cows based on whether they appear on grass rather than learning actual cow features, or identify camels primarily by detecting desert backgrounds [5]. This phenomenon is prevalent even in SSL [11, 22, 29, 10].

While previous research has shown that neural networks are vulnerable to spurious correlations in data [1], several other contributing factors to shortcut learning have been identified. Hermann et al. [17] find shortcuts emerging from color, size, and background. Rahaman et al. [25], Tancik et al. [27] find spectral bias that low-frequency features are learned faster than high-frequency features. While significant progress has been achieved, current theoretical frameworks provide insufficient explanations for why models consistently induce shortcuts.

Recent studies have demonstrated that SSL models with small weight initialization exhibit stepwise learning dynamics, where features are learned sequentially based on the corresponding eigenvalues of the feature cross-correlation matrix [26]. Building on this insight, we analyze the eigenvalue and eigenvector structure of the feature cross-correlation matrix. This approach provides a novel theoretical framework for understanding why certain features, regardless of their semantic importance, are consistently learned earlier in the training process. Our investigation focuses particularly on how dimensional properties influence learning priority, potentially explaining some observed shortcut learning phenomena beyond traditional spurious correlations.

The contributions of our work are as follows:

Submitted to 39th Conference on Neural Information Processing Systems (NeurIPS 2025). Do not distribute.

- We establish theoretical connections between shortcut learning phenomenon, stepwise learning, and eigenvalue-eigenvector of feature cross-correlation matrix on SSL.

- We extend theoretical research on shortcut learning from supervised learning to SSL.

- We characterize *extent bias*, a tendency to prioritize features based on their dimensional extent or spatial coverage rather than their semantic importance.

- We analyze how amplitude and frequency determine which features are learned earlier in SSL, and characterize *amplitude bias*, a tendency to prioritize features based on their amplitude rather than their semantic importance.

## 2 Related Works

**Self-supervised learning** SimCLR [7] established a foundational contrastive learning framework but required large batch sizes to generate sufficient negative pairs for preventing representational collapse. This limitation prompted research into non-contrastive approaches, leading to innovations like SimSiam [8] and BYOL [14]. Further research introduced methods focusing on different training objectives: VICReg [4] introduced variance-invariance-covariance regularization, while Barlow Twins [31] employed cross-correlation matrix to prevent collapse. DINO [6] advanced the field by introducing self-distillation with no labels. The success of DINO v2 [23] sparked interest in Joint Embedding Predictive Architectures (JEPA) [2], with recent work by Littwin et al. [20] revealing JEPA's tendency to prioritize learning "related" features over "frequently" occurring ones.

**Learning dynamics** Following the introduction of Neural Tangent Kernel (NTK) [18], researchers have discovered important connections between eigenvalue dynamics and learning behavior, including spectral bias phenomena [27, 15]. This theoretical framework has enabled deeper analysis of loss function trajectories and saddle point behaviors [19, 24]. Notably, Simon et al. [26] demonstrated that these saddle-to-saddle dynamics appear not only in supervised learning but also extend to SSL settings.

**Shortcut learning** Shortcut learning was first identified in Geirhos et al. [13], describing how neural networks take easier but incorrect paths to solve tasks. This phenomenon appears in various ways: Geirhos et al. [12], Baker et al. [3], Hermann and Lampinen [16] showed that CNNs rely on object texture rather than object shape, Wu et al. [30] demonstrated that even a single pixel can mislead model's decisions, and Hermann et al. [17] revealed that CNNs preferentially learn salient but potentially irrelevant features like scale and background elements. These shortcuts can arise from dataset properties, particularly through spurious correlations [1] and implicit biases. Our work specifically examines how dataset correlations contribute to shortcut learning.

## 3 Background (Stepwise Nature of SSL [26])

In this section, following Simon et al. [26], we analyze the stepwise learning dynamics of SSL systems through the lens of toy Barlow Twins models [31]. We first introduce the loss function and gradient flow dynamics, then derive the connection between cross-correlation matrix and feature learning. Finally, we examine how the eigendecomposition of feature cross-correlation matrix connects to the theoretical foundation for our analysis of extent bias, amplitude bias.

Given training data $\{x^{(i)} \in \mathbb{R}^m : i = 1, 2, \cdots, n\}$, the training loss of toy Barlow twins is defined as $\mathcal{L} = ||C - I_d||_F^2$, $C \equiv \frac{1}{2n} \sum_{i=1}^n (Wx^{(i)})(Wx'^{(i)})^\top + (Wx'^{(i)})(Wx^{(i)})^\top$, where $||\cdot||_F$ is Frobenius norm, $W \in \mathbb{R}^{d \times m}$ is learnable parameters, and $C \in \mathbb{R}^{d \times d}$ is cross-correlation matrix of $Wx$ and $Wx'$ for another view $x'$ from $x$. Using the feature cross-correlation matrix

$$\Gamma \equiv \frac{1}{2n} \sum_{i=1}^n (x^{(i)} x'^{(i)\top} + x'^{(i)} x^{(i)\top}) \in \mathbb{R}^{m \times m}, \tag{1}$$

we have $\mathcal{L} = ||W\Gamma W^\top - I_d||_F^2$ and $C = W\Gamma W^\top$. The eigendecomposition of the feature cross-correlation matrix is $\Gamma = V_\Gamma \Lambda_\Gamma V_\Gamma^\top$ with $\Lambda_\Gamma = \text{diag}(\gamma_1, \cdots, \gamma_m)$ and $V_\Gamma = [v_1 \cdots v_m] \in \mathbb{R}^{m \times m}$, where $\gamma_1 \geq \gamma_2 \geq \cdots \geq \gamma_m$ are eigenvalues of $\Gamma$ and $v_i$'s are the corresponding eigenvectors for $\gamma_i$'s.

Using (3), we can express the gradient flow as follows:

$$\frac{dW}{dt} = -\nabla_W \mathcal{L} = -4(W\Gamma W^\top - I_d)W\Gamma. \tag{2}$$

To analyze eigenvector dynamics of weights, we assume weight initialization is aligned.

**Assumption 3.1** (Aligned Initialization Simon et al. [26])**.** At the initialization, we assume that the right-singular vectors of $W(0)$ are aligned with the top $d$ eigenvectors of $\Gamma$, i.e., the singular value decomposition is $W(0) = US_0V_\Gamma^{(\leq d)\top}$ for a orthogonal matrix $U \in \mathbb{R}^{d \times d}$, the top-$d$ eigenvector matrix $V_\Gamma^{(\leq d)} = [v_1 \cdots v_d] \in \mathbb{R}^{m \times d}$, and a diagonal matrix $S_0 = \text{diag}(s_1(0), \cdots, s_d(0))$ with a small initialization $s_j(0) > 0$.

Under Assumption 3.1, the solution $W(t)$ for the gradient flow (2) can be expressed as follows [26, Proposition 4.1]: $W(t) = US(t)V_\Gamma^{(\leq d)\top}$ for $S(t) = \text{diag}(s_1(t), \cdots, s_d(t))$, where the singular values of $W(t)$ evolve as

$$s_j(t) = \frac{e^{4\gamma_j t}}{\sqrt{s_j^{-2}(0) + (e^{8\gamma_j t} - 1)\gamma_j}}$$

which has a limit of $\gamma_j^{-1/2}$ as $t \to \infty$ and nearly sigmoidal

$$s_j^2(t) \approx \frac{1}{\gamma_j + s_j^{-2}(0)e^{-8\gamma_j t}} =: \tilde{s}_j^2(t). \tag{3}$$

Solving $\tilde{s}_j^2(t) = \frac{1}{2}s_j^2(\infty)$ at its critical time $t = \tau_j$, we have

$$\tau_j = -\frac{\log\left(s_j^2(0)\gamma_j\right)}{8\gamma_j} \tag{4}$$

around which $s_j(t)$ (or $\tilde{s}_j(t)$) passes $\frac{1}{2}\gamma_j^{-1/2}$ and rapidly increases from near zero to near the saturation $\gamma_j^{-1/2}$.

In this paper, we focus on the property that the eigenvector feature $v_j$ corresponding to a larger $\gamma_j$ leads to an earlier critical point $\tau_j$ from (4).

# 4  Extent bias

In computer vision tasks, backgrounds typically span larger regions while foreground objects occupy more concentrated areas. Recent work by Hermann et al. [17] reveals that CNNs preferentially learn these background features over object-specific details, creating a specific form of spurious correlation between backgrounds and class labels. For example, cows are often classified based on grass backgrounds rather than their distinctive features, and camels are identified through desert scenes [5]. This phenomenon points to a underlying learning mechanism we term *extent bias*, a fundamental tendency of neural networks to prioritize features based on their dimensional extent or spatial coverage rather than their semantic importance. The connection between extent bias and learning dynamics implies the need for understanding a more fundamental mechanism beyond traditional spurious correlations. While spurious correlations emerge from dataset-specific relationships, the bias toward learning background features is inherent in the learning dynamics of neural networks themselves. Through our analysis of SSL systems, we demonstrate that this bias for background features emerges naturally from how models learn earlier features with higher extent bias, independent of their semantic relevance or predictive power.

In this section, we investigate how different feature properties influence learning priorities in SSL. Through extent bias analysis, we demonstrate how features with larger dimensional coverage are learned before those with smaller coverage, regardless of their semantic importance.

We construct a theoretical framework that identifies dimensional effects in feature learning. By analyzing how SSL models process features of varying extent bias, we can directly observe how extent bias influences learning priority and connects to the background-foreground learning dynamics observed in practice.

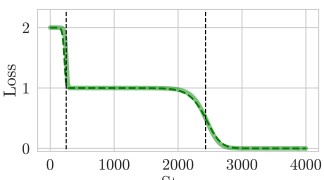 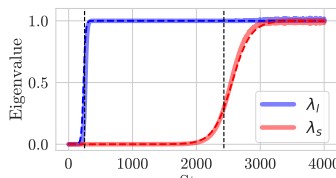 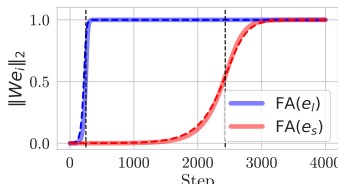

Figure 1: **Effects of extent bias on learning dynamics in SSL.** (Left) Stepwise learning curves of Barlow Twins. There are two ($d = 2$) learning steps shown with two black dashed vertical lines (also shown in the other two panels) which indicate the time steps $t_1$ and $t_2$ with $t_1 : t_2 \approx \frac{1}{\gamma_l} : \frac{1}{\gamma_s} = \frac{1}{m_l} : \frac{1}{m_s}$. The predicted loss (dashed green) of $\mathcal{L} = \sum_{j=1}^d (\tilde{\lambda}_j(t) - 1)^2 = \sum_{j=1}^d (\tilde{s}_j^2(t)\gamma_j - 1)^2$ using (3) match the empirical result (solid green). (Center) Evolution of eigenvalues $\lambda_j$'s of $C$ during training. At the beginning, the first eigenvalue $\lambda_1$ (blue) increases to 1 and then later the second $\lambda_2$ (red) follows. We also compare them with the predicted evolution $\tilde{\lambda}_j(t)$ (dashed lines). (Right) Evolution of the feature alignment $\|We\|_2$ for $e = e_l$ (blue) and $e = e_s$ (red). It shows very similar behaviors with the eigenvalues $\tilde{\lambda}_j^{1/2}$ (dashed lines). See Theorem 4.5. We use $m_l = 9$, $m_s = 1$. See Appendix A.1 for more detailed settings.

## 4.1 Settings

We first consider the following base input $x_{\text{base}} = [b_l \mathbf{1}_{m_l}^\top, b_s \mathbf{1}_{m_s}^\top]^\top \in \mathbb{R}^m$, where $b_l, b_s \overset{\text{i.i.d.}}{\sim} B(p = 0.5)$ follow the Bernoulli distribution and take the value $\pm 1$ with the equal probability, $m_l$ and $m_s$ indicate the size of larger part and smaller part, respectively, i.e., $m_l > m_s$ and $m_l + m_s = m$, and $\mathbf{1}_k$ is the $k$-dimensional all-one vector. From now on, we will use the subscript $l$ and $s$ for the indices with respect to the *larger*-part and *smaller*-part features, respectively.

Then, to obtain the positive pair $(x, x')$, we introduce the following data augmentation $x = x_{\text{base}} + \varepsilon$ and $x' = x_{\text{base}} + \varepsilon'$, with the noise $\varepsilon, \varepsilon' \overset{\text{i.i.d.}}{\sim} \mathcal{N}(0_m, a^2 I_m)$ for some $a > 0$.

## 4.2 Learning Dynamics on extent bias

In this subsection, we discuss the relationship between $\gamma_j$ and $\mathcal{L}$, focusing on which features are learned earlier. From Section 4.1, we can simplify the feature cross-correlation matrix $\Gamma$ by analyzing the expected values of the augmented features. Based on the definition in (1), we have:

$$\Gamma = \frac{1}{2n} \sum_{i=1}^n (x^{(i)} x'^{(i)\top} + x'^{(i)} x^{(i)\top}) = \mathbb{E}[x_{\text{base}} x_{\text{base}}^\top]. \tag{5}$$

To identify which features drive the loss as stepwise phenomena, we consider basis vectors that disentangle individual features. Specifically, we define basis vectors $e_l$ and $e_s$ where each vector has ones only in the dimensions corresponding to its respective feature:

$$e_l = [\mathbf{1}_{m_l}^\top, \mathbf{0}_{m_s}^\top]^\top, e_s = [\mathbf{0}_{m_l}^\top, \mathbf{1}_{m_s}^\top]^\top \in \mathbb{R}^m.$$
$$\text{FA}(e) = \|We\|_2 \text{ for } e = e_l, e_s. \tag{6}$$

By measuring the feature alignment between these basis vectors and the weight matrix through $\text{FA}(e) = \|We\|_2$, we can identify which features are being learned at each stage of the training process.

The eigendecomposition of $\Gamma$ is given by the following proposition:

**Theorem 4.1.** *For the correlation matrix in (5), we have the eigenvalue matrix $\Lambda_\Gamma$ and eigenvector matrix $V_\Gamma$:*

$$\Lambda_\Gamma = diag\left([m_l, m_s, \ \mathbf{0}_{m-2}]\right), V_\Gamma^{(\leq 2)} = [e_l/\sqrt{m_l} \ e_s/\sqrt{m_s}].$$

We defer the proof to Appendix B.1.

We hypothesize that features with larger dimensions are learned faster, regardless of their predictive power or potential to cause shortcuts. This is particularly relevant in vision tasks where such features

might correspond to larger pixel regions. We experiment using a simple toy model to validate our theoretical analysis of dimensional influence on feature learning. In our experimental setup, we used two distinct features with different dimensional coverage ($m_l = 9$ and $m_s = 1$), allowing us to clearly observe the learning dynamics.

As shown in Figure 1, the results demonstrate three key phenomena:

Figure 1 (Left) shows loss trajectory (green line) exhibits two distinct stepwise phenomena, marked by black vertical lines. These stepwise decreases precisely align with the abrupt increase in the eigenvalue observed in Figure 1 (Center), confirming our theoretical prediction that eigenvalue dynamics drives the learning process.

Figure 1 (Center) shows a clear stepwise pattern in which two distinct eigenvalues of $\Gamma$ increase sequentially. This sequential increase directly corresponds to the learning priority of feature, with the higher-dimensional feature ($m_l = 9$) being learned first.

Figure 1 (Right) shows that, feature alignment measurements $||We||_2$ from (6) provide direct evidence of the learning order: the alignment with $e_1$ (blue line, corresponding to the larger feature dimension) increases during the first loss decrease, while $e_2$ alignment (red line) follows during the second phase. This learning pattern strongly supports our hypothesis that dimensional coverage determines how early the features learned.

This result suggests that the spatial extent of features, rather than their semantic content, plays a crucial role in determining learning priority.

### 4.3 Cross-Correlation eigenvalue $\lambda$ and Loss Relationship

In this subsection, we analyze the relationship between the eigenvalues $\lambda_j$ of cross-correlation matrix $C$.

**Theorem 4.2.** *Under Assumption 3.1, the eigenvalues $\lambda_j$ of feature cross-correlation matrix $C = W\Gamma W^\top$, using the approximation $s_j \approx \tilde{s}_j$ in (3), are approximated as $\lambda_j = s_j^2 \gamma_j \approx \tilde{s}_j^2 \gamma_j =: \tilde{\lambda}_j$ which have*

$$\tilde{\lambda}_j(\tau_j) = \frac{1}{2} \text{ and } \tilde{\lambda}_i'(\tau_j) \begin{cases} = 2\gamma_j & \text{if } i = j, \\ \approx 0 & \text{if } i \neq j \end{cases} \tag{7}$$

*at $\tau_j = -\log(s_j^2(0)\gamma_j)/8\gamma_j$ in (4). For the Barlow Twins loss $\mathcal{L} = \|C - I_d\|_F^2$, we have $\mathcal{L} = \sum_{j=1}^d (\lambda_j - 1)^2$ and $-\frac{d\mathcal{L}}{dt}(\tau_j) \approx \tilde{\lambda}_j'(\tau_j) = 2\gamma_j$.*

We defer the proof to Appendix B.3.

Figure 6 in Appendix C shows the relationship between cross-correlation eigenvalue $\lambda$ differentiated with respect to $t$ and loss derivatives $\frac{d\mathcal{L}}{dt}$. The close alignment between the loss derivative and $\lambda$ derivative curves demonstrates that the decrease in loss is directly driven by $\lambda$, with larger $m_l$ features learned, and smaller $m_s$ features learned later. The curves' relative magnitudes show an approximate $m_l : m_s$ ratio, which matches our theoretical predictions.

### 4.4 Weight Singular Value Evolution

To verify the dynamics of weight singular values $s_j$, we propose the following theorem:

**Theorem 4.3.** *Using the approximation (3), the singular values of the weight matrix $W$ satisfy*

$$\tilde{s}_j(\tau_j) = 1/\sqrt{2\gamma_j} \text{ and } \tilde{s}_j'(\tau_j) = \sqrt{2\gamma_j}$$

*at the critical point $t = \tau_j$.*

We defer the proof to Appendix B.4.

Figure 7 in Appendix C shows two key aspects of singular value dynamics during training. First, the singular values $s_j$ evolve to their theoretical limits $1/\sqrt{\gamma_j}$ and $1/\sqrt{\gamma_s}$, as predicted by our analysis. Second, the derivatives of these singular values exhibit peaks at their respective critical points, with magnitudes that follow the predicted $\sqrt{2\gamma_l} : \sqrt{2\gamma_s}$ ratio. These results provide strong empirical validation of our theoretical framework, demonstrating that both the convergence values and learning priority on different features are governed by their corresponding eigenvalues in the feature cross-correlation matrix $\Gamma$.

## 4.5 Aligned Initialization and Subspace Alignment

To justify our alignment initialization assumption in Assumption 3.1, we first define the following subspace alignment metric:

**Definition 4.4** (Subspace Alignment). We define subspace alignment of two subspaces $\text{Im}(A)$ and $\text{Im}(B)$:

$$\text{SA}(A, B) = ||A^\top B||_F^2 / d,$$

where $\text{Im}(A) = \{Av \in \mathbb{R}^m : v \in \mathbb{R}^d\}$, $A = [a_1 \cdots a_d], B = [b_1 \cdots b_d] \in \mathbb{R}^{m \times d}$ and $a_i, b_i \in \mathbb{R}^m$ are unit vectors.

Note that $0 \leq \text{SA}(A, B) \leq 1$ and it attains $\text{SA}(A, B) = 0$ when $\text{Im}(A) \perp \text{Im}(B)$, and $\text{SA}(A, B) = 1$ when $\text{Im}(A) = \text{Im}(B)$. Figure 10 (Top) in Appendix D empirically validates Assumption 3.1 using the subspace alignment metric. The model becomes aligned rapidly in the early stages of training, satisfying the assumption.

## 4.6 Orthogonal Feature Learning

Our analysis shows that features are learned as orthogonal to each other, where each feature is acquired independently without interference from others. This orthogonal learning pattern is particularly evident in the evolution of the model's weight matrix singular vectors. To formalize this observation, we analyze how the left singular vectors of the weight matrix align with the feature vectors during training.

**Theorem 4.5.** *Under Assumption 3.1, the left singular vectors $u$ of $W(t)$ learn features orthogonally:*

$$Proj_{U(\leq 2)}(We_l) := (u_l^\top We_l, u_s^\top We_l) = (\sqrt{\lambda_l}, 0),$$
$$Proj_{U(\leq 2)}(We_s) := (u_l^\top We_s, u_s^\top We_s) = (0, \sqrt{\lambda_s}),$$

*where $u_l, u_s$ are the corresponding left singular vectors for the singular values $s_l, s_s$.*

Figure 11 shows orthogonal learning pattern that features are learned independently and sequentially, supporting our theoretical analysis of stepwise learning dynamics.

## 4.7 Non-linear multi layer network

Nonlinearity exhibits distinct learning dynamics compared to linearity. Therefore, we aim to investigate whether extent biass also exists in multilayer perceptrons (MLPs). We experiment with a 3-layer network, using leakyReLU as the activation function, for understanding non-linear feature learning dynamics. Our non-linear network experiments demonstrate that extent bias persists beyond linear models. As shown in Figure 14 in Appendix G, the non-linear network exhibits remarkably similar stepwise learning patterns to those observed in linear models Figure 1. Key similarities include: similar eigenvalue evolution patterns, consistent stepwise loss reduction phases. These results suggest that extent bias is a fundamental learning phenomenon that transcends network architecture complexity, rather than being merely an artifact of linear models.

## 4.8 Practical Study on Colored-MNIST Dataset

We conducted experiments using a Colored-MNIST dataset, where we adjusted the ratio of digits pixels relative to the total image pixels. We tested three different ratios: 0.05, 0.10, and 0.15. In this dataset, we set the correlation between background and label to 70% for both training and test sets, making it difficult for a model that predicts solely based on background to achieve accuracy higher than 70%. According to our hypothesis, since backgrounds have larger extent bias than objects, the test set accuracy would rapidly increase from an initial 10% (random choosing) to 70% (as the model learns background features), then plateau for a period, before slowly rising to 100% (as it learns object features). We also hypothesized that this plateau period would decrease as the ratio of label pixels increases in the images, with shorter plateaus observed in the 0.15 ratio condition compared to 0.05.

Figure 2 supports our hypothesis. Across all pixel ratio conditions (0.05, 0.10, 0.15), test accuracy exhibited a consistent pattern: a rapid increase from initial 10% to 70%, followed by a plateau period,

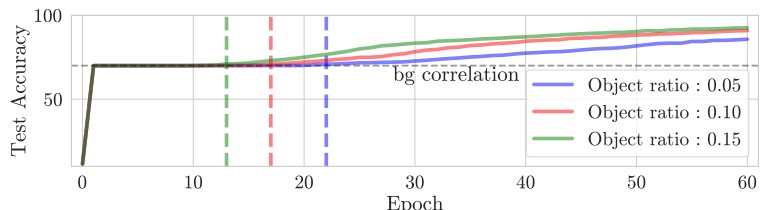

Figure 2: **Extent bias effects on spurious datasets.** ResNet18 on the Colored MNIST dataset. (Left) Loss decreases even though the error rate doesn't decrease. (Right) The error rate has a plateau at 70%, which corresponds to the correlation between background and object. The lengths of the plateaus become shorter as the object's pixel ratio increases. See Appendix A.2 for more detailed settings.

and then a gradual ascent to 100%. Notably, as the object pixel ratio increased, the duration of the plateau phase decreased. The loss function continued to decrease even when accuracy remained stagnant at 70%. This suggests a extent bias where larger objects are prioritized during the learning process. The pattern reflects how the model initially achieves 70% accuracy by relying on background features, which statistically occupy larger regions, before progressively learning object features. Furthermore, this indicates that larger extents occupy greater eigenvalues, implying a reduction in the critical point $\tau_j$.

## 5   Amplitude Bias

In regression tasks, the phenomenon of spectral bias has been observed, wherein low-frequency components are learned more rapidly than high-frequency components during the training process. Conversely, in classification tasks, a phenomenon known as frequency shortcut [28] has been observed, wherein the model preferentially learns the distinctive Fourier components of the input during the training process. While these studies have primarily focused on supervised learning, we extend this investigation to the SSL, seeking to understand whether similar learning dynamics persist within SSL frameworks.

### 5.1   Settings

To analyze how frequency and amplitude bias affect learning dynamics, we consider input data $x_{\text{base}} \in \mathbb{R}^m$ composed of two sinusoidal components with different frequencies:

$$x_{\text{base}}[t] = c_h b_h \sin(f_h t) + c_l b_l \sin(f_l t), \tag{8}$$

where $f_h = \frac{2\pi}{m} k$ and $f_l = \frac{2\pi}{m} k'$ represent different frequencies for some integers $k$ and $k'$, $b_h, b_l \overset{\text{i.i.d.}}{\sim} B(p = 0.5)$ follow the Bernoulli distribution and take the value $\pm 1$. Suppose $f_h < f_l$ to examine the learning dynamics between low and high frequency components. The coefficients $c_h$ and $c_l$ control the amplitude of each sinusoidal component, allowing us to investigate how magnitudes affect learning earlier. The Bernoulli variables $b_h$ and $b_l$ introduce phase reversal in the signal. The time vector $t$ spans the input dimension $m$. We use the same augmentation with (4.1) to generate positive pairs $(x, x')$ by adding Gaussian noise.

### 5.2   Learning Dynamics on Amplitude Bias

Similar to Section 4.2, we consider basis vectors $e_h$ and $e_l$ that isolate individual features: $e_h = c_h \sin(f_h t)$ and $e_l = c_l \sin(f_l t)$, where $0 \leq t \leq m$. Note that these two are orthogonal since $f_h = \frac{2\pi}{m} k$ and $f_l = \frac{2\pi}{m} k'$ with $k \neq k'$. Similar to Theorem 4.1, the cross-correlation matrix $\Gamma$ for the data generated from (8) can be expressed as follows:

**Theorem 5.1.** *Under (8), the correlation matrix $\Gamma$ has*

$$\Lambda_\Gamma = diag\left(\left[c_h^2 m/2, c_l^2 m/2, \mathbf{0}_{m-2}\right]\right), V_\Gamma^{(\leq 2)} = [e_h \ e_l].$$

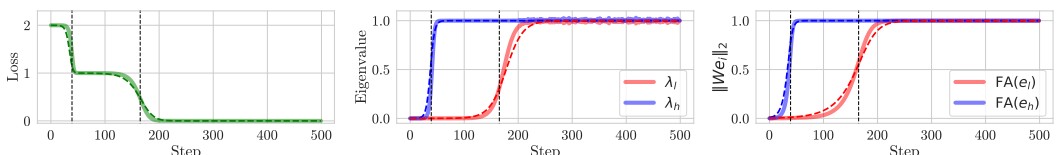

Figure 3: **Amplitude bias effects on learning dynamics in SSL.** See the caption of Figure 1. Note that the time steps $t_1$ and $t_2$ with $t_1 : t_2 \approx \frac{1}{\gamma_h} : \frac{1}{\gamma_l} = \frac{1}{c_h^2} : \frac{1}{c_l^2}$. We use $c_h = 1$, $c_l = 1/2$. See Appendix A.3 for more detailed settings.

We defer the proof to Appendix B.2.

From (9), we observe that eigenvalues are proportional to the squares of the coefficients $c_h^2$ and $c_l^2$. This implies that the learning dynamics are more strongly influenced by the amplitude rather than the underlying frequency.

To validate our theoretical analysis of amplitude bias effect on learning dynamics, we conduct experiments using input data defined in (8). Especially, we set $c_h > c_l$. This configuration shown in Figure 4 in Appendix A, allows us to examine how high-amplitude $c_h \sin(f_h t)$ and low-amplitude $c_l \sin(f_l t)$ affects feature amplitude bias. More details about the experiment are in Appendix A.3.

Our analysis reveals two dominant eigenvalues. The large eigenvalue corresponds to the high-amplitude feature, and small eigenvalue corresponds to the low-amplitude component. The eigenvectors of $\Gamma$ are shown in Figure 5 , Appendix A. The first eigenvector, which corresponds to the largest eigenvalue, captures the dominant high-amplitude oscillation. The second eigenvector, which matches next-largest eigenvalue, captures the low-amplitude oscillation. Other eigenvectors are noise, corresponding to eigenvalues that are almost 0.

### 5.3 Cross-Correlation eigenvalue $\lambda$ and Loss Relationship

We analyze how the eigenvalues $\lambda$ relate to the loss dynamics. The relationship follows similar patterns to those observed in Section 4.3, but with coefficients $c_h$ and $c_l$ rather than $m_l$ and $m_s$.

Figure 8 in Appendix C shows the close relationship between the derivatives of cross-correlation eigenvalues $\frac{d\lambda_h}{dt}$, $\frac{d\lambda_l}{dt}$ and $\frac{d\mathcal{L}}{dt}$. The peaks in these derivatives occur at the critical points with magnitudes proportional to the corresponding coefficients $\gamma_h : \gamma_l = c_h^2 : c_l^2$ (see (9)). This shows our theoretical predictions Theorem 4.2 matches empirical result.

### 5.4 Weight Singular Value Evolution

We now analyze how the singular values of the weight matrix evolve during training. Similarly to the extent bias case, we expect the singular values $s_j$ to converge to theoretical limits determined by the feature coefficients.

Figure 9 in Appendix C shows the evolution of singular values $s_h$ and $s_l$ of weight matrix $W$ (Left) and their derivatives (Right). The singular values converge to their theoretical limits $1/\sqrt{\gamma_j}$ predicted by Theorem 4.3, where $\gamma_j = c_j^2 \frac{m}{2}$. At the critical points $\tau_j$, the derivatives achieve their maximum values of $\sqrt{2\gamma_j}$, showing that rates of feature learning are proportional to the coefficients. These results confirm that the feature coefficients, rather than their frequencies, govern both the convergence values and rates of feature learning.

### 5.5 Aligned Initialization and Subspace Alignment

To validate Assumption 3.1 about alignment between the weight matrix singular vectors and eigenvectors of $\Gamma$, we measure the subspace alignment metric as defined in the extent case Definition 4.4. Figure 10 (Bottom) in Appendix D empirically validates our assumption through subspace alignment measurements. As discussed in Section 4.5, the model achieves alignment rapidly in the early stages of training, even with small random initializations.

### 5.6 Orthogonal Feature Learning

Similar to the extent case, we investigate how the weight matrix learns different frequency components orthogonally as shown in Theorem 4.5. The orthogonal learning pattern reveals how frequency features are acquired independently despite their different spectral characteristics.

Figure 12 in Appendix E shows the trajectories of weight matrix in terms of their alignments with frequency components $e_h$ and $e_l$. The blue trajectory shows the first learning phase where $u_1$ aligns with the high-amplitude feature ($c_h \sin(f_h t)$), followed by the red trajectory showing $u_2$ aligning with the low-amplitude feature ($c_l \sin(f_l t)$). This sequential, orthogonal learning pattern demonstrates that feature learning is primarily determined by coefficient magnitudes rather than frequency characteristics, supporting our analysis in Theorem 4.5.

### 5.7 Non-linear multi layer network

Same as Section 4.7 in Appendix G, we conduct experiments with a 3-layer network using leakyReLU activations to analyze how amplitude coefficients affect learning dynamics in non-linear settings.

Figure 15 in Appendix G demonstrates amplitude bias effects in non-linear networks is similar to linear networks on Figure 3. These results confirm that amplitude bias persists in non-linear architectures, suggesting amplitude magnitude remains a primary determinant of feature learning priority regardless of network complexity.

### 5.8 Discussion

Figure 13 in Appendix F shows that a learning process is driven primarily by feature coefficient magnitude rather than frequency characteristics. The key observation is that the first learned features are those with large coefficients, independent of their spectral properties. This finding parallels frequency shortcut [28] in classification tasks, but reveals a different underlying mechanism. While frequency shortcut suggests models preferentially learn distinctive Fourier components, our results demonstrate that amplitude magnitude—not frequency characteristics—primarily determines feature learning priority.

## 6 Conclusion

In this work, we establish a theoretical connection between eigendecomposition of the feature cross-correlation matrix, shortcut learning, and stepwise learning behavior in SSL. We provide insights into how dimensional feature properties influence the learning process in SSL frameworks. This work not only explains observed shortcut learning phenomena but also offers a theoretical lens for understanding and potentially mitigating such learning biases. This theoretical framework lays the groundwork for developing more robust SSL algorithms. Future work should focus on leveraging these insights to design mechanisms that encourage learning of generalizable features despite their potentially lower extent bias or amplitude bias.

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

# A  Experimental Details

## A.1  Extent bias Experiment

For the extent bias experiment shown in Section 4.1, we train the model using 400 epochs. The augmentation noise parameter $a$ was set to 0.01. We use a dataset size of $n = 1000$ samples with feature dimension $m = 10$. We also use learning rate $\eta = 6 \cdot 10^{-4}$ and scaling factor $5 \cdot 10^{-1}$.

## A.2  Colored MNIST Experiment

For the Colored MNIST shown in Section 4.8, we train the model using default augmentation (RandomResizedCrop, RandomHorizontalFlip, RandomColorJitter, RandomGrayscale, Random-GaussianBlur, RandomSolarization) with augmentated image size $42 \times 42$. We use background colors as [[255, 0, 0], [0, 255, 0], [0, 0, 255], [255, 255, 0], [255, 0, 255], [0, 255, 255], [0, 123, 123], [123, 0, 123], [123, 123, 0], [123, 0, 0]][digit]. We trained ResNet18 with 60 epochs, AdamW [21] with learning rate $\eta = 4 \times 10^{-6}$.

## A.3  Amplitude Experiment

For the amplitude experiment shown in Section 5.1, we train the model using 500 epochs. The augmentation noise parameter $a$ is set to 0.1. We use a dataset size of $n = 1000$ samples with feature frequency $f_h = 2\frac{2\pi}{24}, f_l = 32\frac{2\pi}{24}$. We also use learning rate $\eta = 5 \cdot 10^{-5}$, scaling factor $3 \cdot 10^{-3}$ and $m = 96$.

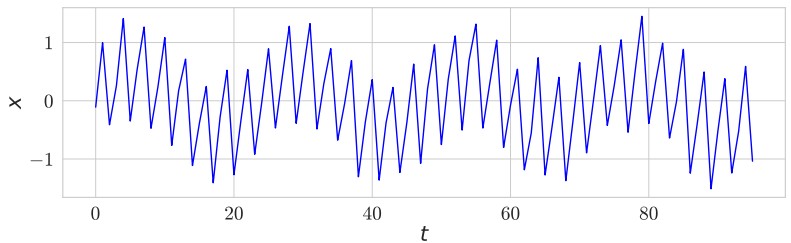

Figure 4: **Input data** $x = x_{base} + \epsilon$. $x_{\text{base}}[t] = b_h c_h \sin(f_h t) + b_l c_l \sin(f_l t)$, where $c_h = 1, c_l = 0.5, f_h = \frac{2\pi}{m}32, f_l = \frac{2\pi}{m}8, m = 96$.

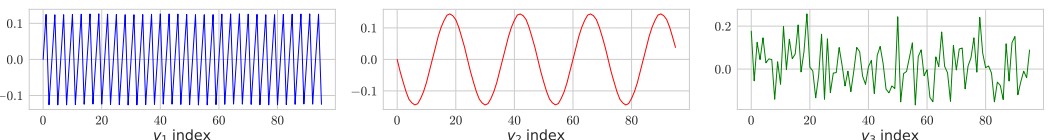

Figure 5: **The eigenvectors $v_i$'s of $\Gamma$ for $i = 1, 2, 3$ (from Left to Right).** (Left) The first eigenvector that correspondent to the largest eigenvalue indicates the (high frequency) feature with a high amplitude $c_h \sin(f_h t)$, (Center) the second the (low frequency) feature with a low amplitude feature $c_l \sin(f_l t)$, (Right) the third (and beyond) noise, where $c_l < c_h$.

# B  Proofs

## B.1  Proof of Theorem 4.1

Through matrix analysis, we can express:

$$\Gamma = \mathbb{E}[x_{\text{base}}x_{\text{base}}^\top] = \begin{bmatrix} \mathbf{1}_{m_l \times m_l} & \mathbf{0}_{m_s \times m_l} \\ \mathbf{0}_{m_l \times m_s} & \mathbf{1}_{m_s \times m_s} \end{bmatrix},$$

438 which has two eigenvectors $e_l/\|e_l\|$ and $e_s/\|e_s\|$ correspond to nonzero eigenvalues. We get the
439 eigenvalues $m_l$ and $m_s$ from the following equation:

$$\det(\Gamma - \lambda I) = \det(\mathbf{1}_{m_l \times m_l} - \lambda I_{m_l \times m_l}) \det(\mathbf{1}_{m_s \times m_s} - \lambda I_{m_s \times m_s}) = 0.$$

440 Finally, we can get the eigendecomposition $\Gamma = V_\Gamma \Lambda_\Gamma V_\Gamma$ where

$$\Lambda_\Gamma = \operatorname{diag}\left([m_l, m_s, \, \mathbf{0}_{m-2}]\right),$$

$$V_\Gamma^{(\le d)} = \left[ \frac{1}{\sqrt{m_l}} e_l \quad \frac{1}{\sqrt{m_s}} e_s \right].$$

## B.2 Proof of Theorem 5.1

442 The cross-correlation matrix $\Gamma$ for this input can be expressed using (5):

$$
\begin{aligned}
\Gamma &= \mathbb{E}[x_{\text{base}} x_{\text{base}}^\top] \\
&= \mathbb{E}[c_h^2 b_h^2 \sin(f_h t) \sin(f_h t)^\top + c_l^2 b_h^2 \sin(f_l t) \sin(f_l t)^\top + c_h c_l b_h b_l \sin(f_h t) \sin(f_l t)^\top + c_h c_l b_h b_l \sin(f_l t) \sin(f_h t)^\top] \\
&= c_h^2 \sin(f_h t) \sin(f_h t)^\top + c_l^2 \sin(f_l t) \sin(f_l t)^\top.
\end{aligned}
$$

443 Using the orthogonality between $\sin(f_h t)$ and $\sin(f_l t)$ ($f_h \ne f_l$), where $t \in \mathbb{N}$,

$$
\begin{aligned}
\Gamma &= c_h^2 \sin(f_h t) \sin(f_h t)^\top + c_l^2 \sin(f_l t) \sin(f_l t)^\top, \\
\Gamma \sin(f_h t) &= c_h^2 \| \sin(f_h t) \|^2 \sin(f_h t), \\
\Gamma \sin(f_l t) &= c_l^2 \| \sin(f_l t) \|^2 \sin(f_l t).
\end{aligned}
$$

444 We find eigenvector and eigenvalue as:

$$\Lambda_\Gamma = \operatorname{diag}\left( \left[ c_h^2 \| \sin(f_h t) \|^2, c_l^2 \| \sin(f_l t) \|^2, \mathbf{0}_{m-2} \right] \right),$$

$$V_\Gamma^{(\le 2)} = [e_h \ e_l]^\top.$$

445 With $f = \frac{2\pi}{m} k$ for some integer $k$, we have

$$
\begin{aligned}
\| \sin(fx) \|^2 &= \int_0^m \sin^2(fx) dx = \int_0^m \frac{1 - \cos(2fx)}{2} dx \\
&= \frac{1}{2} \left[ x - \frac{\sin(2fx)}{2} \right]_0^m = \frac{m}{2} - \frac{\sin(2fm)}{4} = \frac{m}{2}.
\end{aligned}
$$

446 Finally, we have

$$\Lambda_\Gamma = \operatorname{diag}\left( \left[ c_h^2 \frac{m}{2}, c_l^2 \frac{m}{2}, \mathbf{0}_{m-2} \right] \right),$$

$$V_\Gamma^{(\le 2)} = [e_h \ e_l].$$

## B.3 Proof of Theorem 4.2

448 We have

$$\tilde{\lambda}_j(t) = \tilde{s}_j^2(t) \gamma_j = (1 + \lambda_j(0)^{-1} e^{-8\gamma_j t})^{-1},$$

449 and thus if we plug in $\tau_j = -\log(\lambda_j(0))/8\gamma_j$, i.e., $\exp(-8\gamma_j \tau_j) = \lambda_j(0)$, then we have $\tilde{\lambda}_j(\tau_j) =$
450 $(1 + 1)^{-1} = \frac{1}{2}$. The derivative $\tilde{\lambda}_j'(t)$ at $t = \tau_j$ is given as follows:

$$
\begin{aligned}
\tilde{\lambda}_j'(t) &= -(1 + \lambda_j(0)^{-1} e^{-8\gamma_j t})^{-2} (-8\gamma_j \lambda_j(0)^{-1} e^{-8\gamma_j t}) \\
&= -\tilde{\lambda}_j^2(t) (-8\gamma_j \lambda_j(0)^{-1} e^{-8\gamma_j t}) \\
\tilde{\lambda}_j'(\tau_j) &= -\tilde{\lambda}_j^2(\tau_j) (-8\gamma_j \lambda_j^{-1}(0) \lambda_j(0)) \\
&= 2\gamma_j.
\end{aligned}
$$

Using the equations

$$C = \sum_{j=1}^{d} \lambda_j u_j u_j^\top \text{ and } C^2 = \sum_{j=1}^{d} \lambda_j^2 u_j u_j^\top,$$

we get the loss

$$\mathcal{L} = ||C - I||_F^2 = \text{Tr}((C - I)(C - I)) = \text{Tr}(C^2) - 2\,\text{Tr}(C) + d$$

$$= \sum_{j=1}^{d} \lambda_j^2 - 2\sum_{j=1}^{d} \lambda_j + d = \sum_{j=1}^{d} (\lambda_j - 1)^2.$$

Thus, we get the following equation:

$$\frac{d\mathcal{L}}{dt}(\tau_j) = \sum_{i=1}^{d} 2(\lambda_i(\tau_j) - 1)\lambda_i'(\tau_j)$$

$$\approx \sum_{i=1}^{d} 2(\tilde{\lambda}_i(\tau_j) - 1)\tilde{\lambda}_i'(\tau_j)$$

$$\approx 2(\tilde{\lambda}_j(\tau_j) - 1)\tilde{\lambda}_j'(\tau_j)$$

$$= -\tilde{\lambda}_j'(\tau_j) = -2\gamma_j.$$

## B.4 Proof of Theorem 4.3

First, we have

$$\tilde{s}_j(t) = (\gamma_j + s_j^{-2}(0)\exp(-8\gamma_j t))^{-1/2},$$

$$\tilde{s}_j(\tau_j) = (\gamma_j + s_j^{-2}(0)\lambda_j(0))^{-1/2}$$

$$= (2\gamma_j)^{-1/2}.$$

and its derivative is given as follows:

$$\tilde{s}_j'(t) = -\frac{1}{2}(\gamma_j + s_j^{-2}(0)\exp(-8\gamma_j t))^{-3/2}(-8\gamma_j s_j^{-2}(0)\exp(-8\gamma_j t)),$$

$$\tilde{s}_j'(\tau_j) = -\frac{1}{2}(\gamma_j + s_j^{-2}(0)\lambda_j(0))^{-3/2}(-8\gamma_j s_j^{-2}(0)\lambda_j(0))$$

$$= -\frac{1}{2}(2\gamma_j)^{-3/2}(-8\gamma_j^2)$$

$$= (2\gamma_j)^{1/2}.$$

 # C   Derivatives

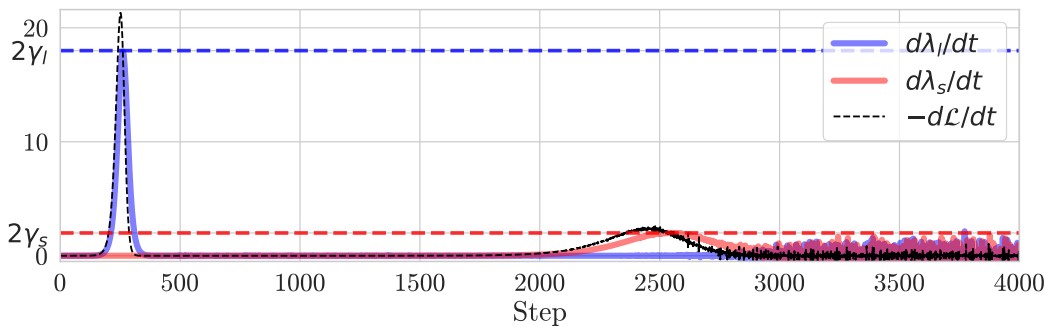

Figure 6: **Derivatives** $\frac{d\lambda_l}{dt}$ **(blue),** $\frac{d\lambda_s}{dt}$ **(red), and** $-\frac{d\mathcal{L}}{dt}$ **(black dashed).** The derivative $\frac{d\lambda_l}{dt}(\tau_l)$ (solid blue), $\frac{d\lambda_s}{dt}(\tau_s)$ (solid red) are approximately equal to $2\gamma_l = 2m_l$ (dashed blue), $2\gamma_s = 2m_s$ (dashed red).

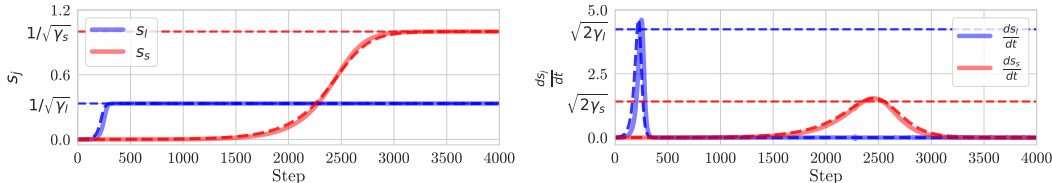

Figure 7: **Evolution of** $s_j(t)$ **and** $s_j'(t)$**.** (Left) Evolution of singular values $s_l$ (solid blue) and $s_s$ (solid red) of $W$ during training. They converge near to $1/\sqrt{\gamma_l} = 1/3$ (dashed horizontal blue) and $1/\sqrt{\gamma_s} = 1$ (dashed horizontal red), respectively. The predicted singular values (dashed blue, dashed red) match the empirical result. (Right) Evolution of the derivatives $\frac{ds_l}{dt}$ (solid blue) and $\frac{ds_s}{dt}$ (solid red). The derivatives $\frac{ds_l}{dt}(\tau_l)$, $\frac{ds_s}{dt}(\tau_s)$ are approximately equal to $\sqrt{2\gamma_l}$ (dashed horizontal blue), $\sqrt{2\gamma_s}$ (dashed horizontal red). The predicted derivatives of singular values (dashed blue, dashed red) also match the empirical result. We use $m_l = 9$ and $m_s = 1$.

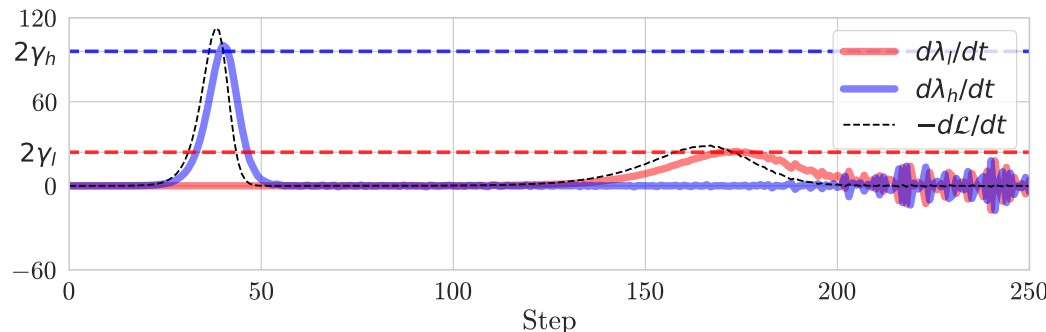

Figure 8: **Derivatives** $\frac{d\lambda_h}{dt}$ **(blue),** $\frac{d\lambda_l}{dt}$ **(red), and** $-\frac{d\mathcal{L}}{dt}$ **(black dashed).** The derivative $\frac{d\lambda_h}{dt}(\tau_h)$ (solid blue), $\frac{d\lambda_l}{dt}(\tau_l)$ (solid red) are approximately equal to $2\gamma_h = 2c_h^2$(dashed blue), $2\gamma_l = 2c_l^2$(dashed red). See Figure 6 together.

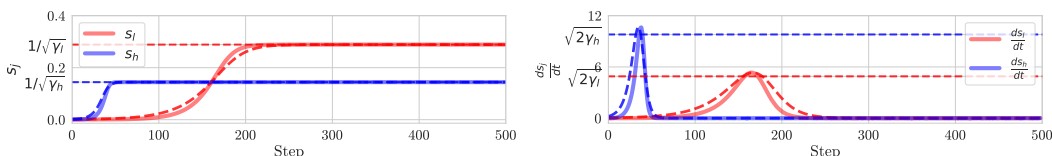

Figure 9: **Evolution of** $s_j(t)$ **and** $s_j'(t)$**.** See the caption of Figure 7. (Left) They converge near to $1/\sqrt{\gamma_h} = 1/\sqrt{c_h^2 \frac{m}{2}}$ and $1/\sqrt{\gamma_l} = 1/\sqrt{c_l^2 \frac{m}{2}}$ . (Right) The derivatives $\frac{ds_h}{dt}(\tau_h)$, $\frac{ds_l}{dt}(\tau_l)$ are approximately equal to $\sqrt{2\gamma_h}$, $\sqrt{2\gamma_l}$. We use $c_h = 1$ and $c_l = 1/2$.

# D   Subspace Alignment

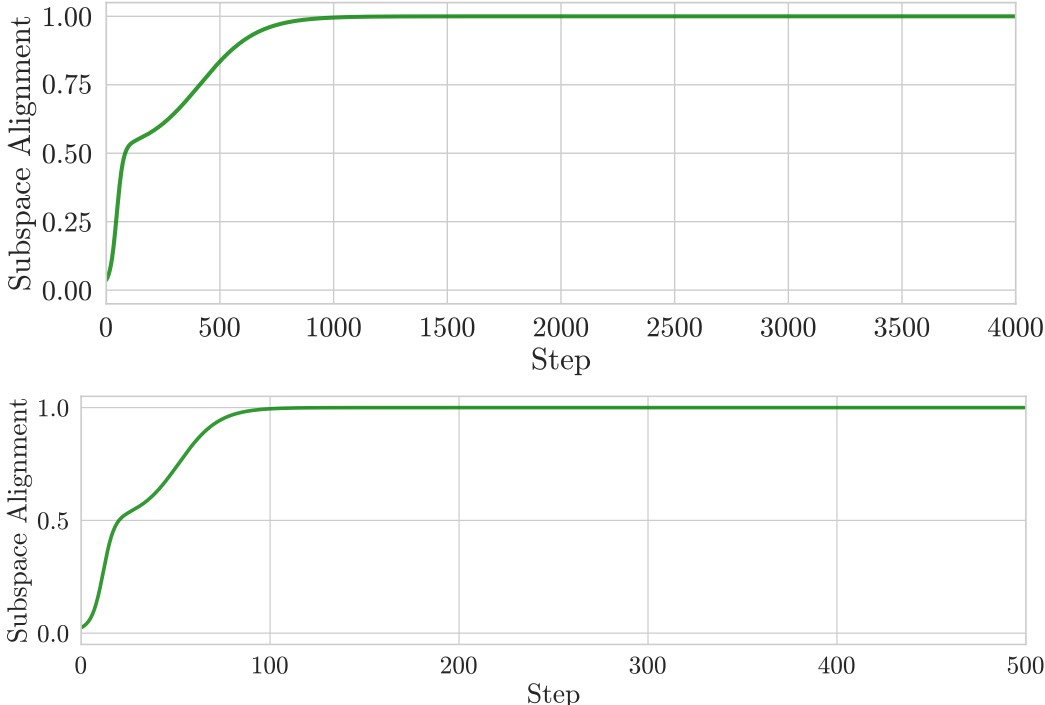

Figure 10: Evolution of subspace alignment $\text{SA}(V^{(\leq d)}, V_\Gamma^{(\leq d)})$ ($d = 2$) between the top-$d$ right singular vectors of $W$ and eigenvectors of $\Gamma$. We use the data (Top) from Section 4.1 and (Bottom) from Section 5.1. See Appendix A.

 # E Orthogonal Feature Learning

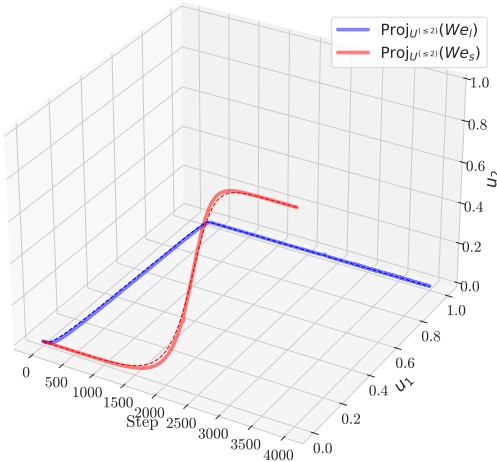

Figure 11: **Visualization of the trajectory of $We_l$ and $We_s$ on the subspace spanned by $u_1, u_2$ during training.** The high-dimensional feature $We_h$ (blue solid line) aligns with $u_1$ and the low-dimensional feature $We_l$ (red solid line) aligns with $u_2$. Dashed lines are predicted trajectory (see Theorem 4.5).

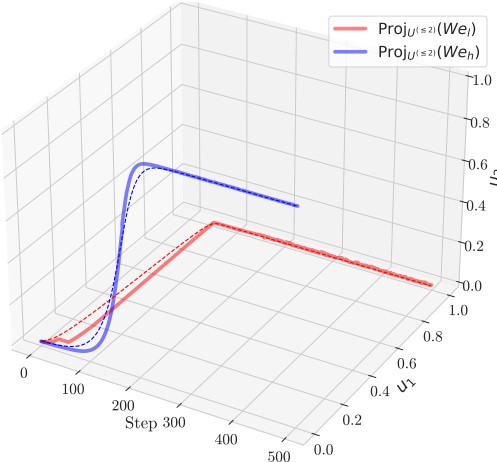

Figure 12: **Visualization of the trajectory of $We_h$ and $We_l$ on the subspace spanned by $u_1, u_2$ during training.** See the caption of Figure 11.

 # F    Right Singular Vectors of $W$

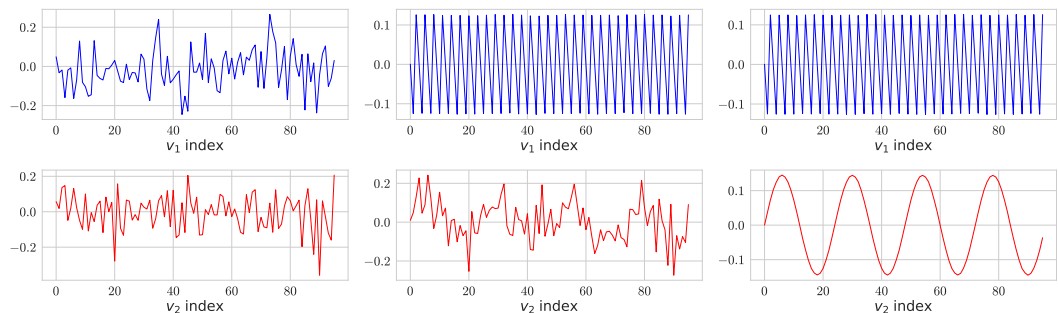

Figure 13: **The first two right singular vectors (Top/Bottom) of $W$ during training (from Left to Right).** (Left) At $t = 0$, the two singular vectors are just noise. (Center) A little after $t = \tau_1$, the first singular value reaches the plateau as shown in Figure 3 and only the (high frequency) feature with a high amplitude is learned. (Right) At the convergence, the model learns the two features.

 # G   Non-linear Experiments

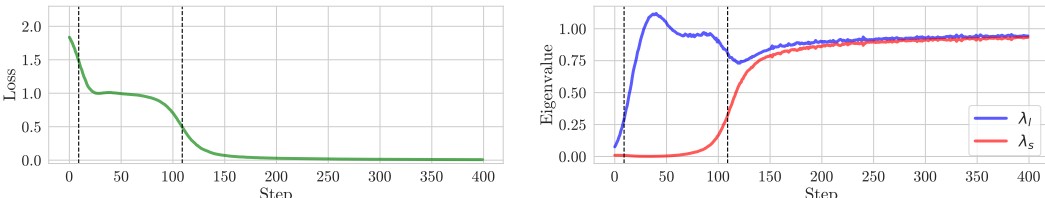

Figure 14: **Effects of extent bias on learning dynamics in non-linear network.** (Left) Stepwise learning curves of Barlow Twins. There are two ($d = 2$) learning steps shown with two black dashed vertical lines (also shown in the other two panels) on empirical result (solid green). (Right) Evolution of eigenvalues $\lambda_j$'s of $C$ during training. At the beginning, the first eigenvalue $\lambda_1$ (blue) increases to 1 and then later the second $\lambda_2$ (red) follows. We use same inputs in Figure 1.

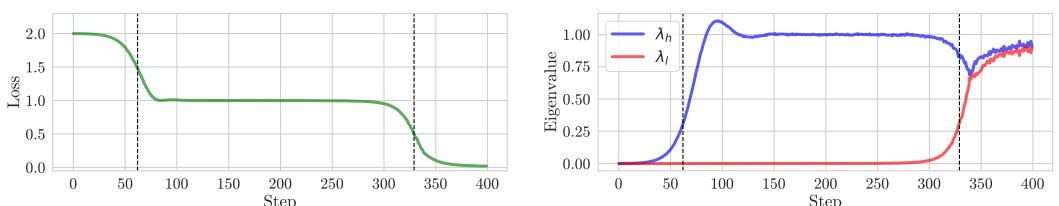

Figure 15: **Amplitude bias effects on learning dynamics in non-linear network.** (Left) Stepwise learning curves of Barlow Twins showing two distinct learning phases with vertical dashed lines marking critical transition points during training. The green line shows empirical loss decreasing in two clear stages. (Right) Evolution of eigenvalues $\lambda_j$ of correlation matrix C during training. The eigenvalue $\lambda_l$ (blue) increases first, followed by the eigenvalue $\lambda_s$ (red), demonstrating amplitude-based learning prioritization. We use same inputs in Figure 3.

## H Limitations

Our study has several limitations due to its simplified assumptions. While our theoretical analysis provides valuable insights into the relationship between extent bias and shortcut learning, several limitations should be acknowledged:

- Linear Network Assumption: We focus on one-layer linear networks, which may not capture the complexities of multi-layer non-linear neural networks.

- Feature Independence: Our assumption of independent features may not reflect the complex interdependencies in practical scenarios.

- Augmentation Limitations: Augmentation Limitations: Our basic augmentation approach may not fully represent the sophisticated strategies used in modern SSL methods.

Future work could address these limitations by extending the theoretical framework to non-linear networks, incorporating feature interactions, and analyzing the impact of more complex augmentation strategies.

## I Supplementary Studies

### I.1 Non-linear Feature Learned Measurement

Nonlinearity exhibits distinct learning dynamics compared to linearity. Therefore, we aim to investigate whether extent biases also exists in multilayer perceptrons (MLPs). We define a measurement of feature learning as:

**Definition I.1.** (Feature Learning Distance). When a model $f(\cdot, \theta)$ has sufficiently learned a specific latent feature vector $e_f$, $f(X, \theta)$ contains information about $e_f$ for input $X = p(e_f) \in R^m$ where $p$ represents some non-linear transformation function. Consequently, if a simple linear probing function $g$ can extract $e_f$ from $f(X, \theta)$, we can define that the model $f$ has meaningfully learned $e_f$. Furthermore, to quantify the degree of learning, assuming an optimally trained probe $g$, we define a feature learning metric

$$\text{FLD}(k) = \min_g \mathbb{E}_{e_f \in \mathcal{P}_k} \left[ \frac{\text{MSE}(g(f(X, \theta)), e_f)}{||e_f||_2^2} \right], \tag{9}$$

where $\mathcal{P}_k$ is distribution of feature $k$.

### I.2 Non-linear on extent bias

We experiment on Section 4.7, for understanding non-linear feature learning dynamics. Figure 14 shows this results.

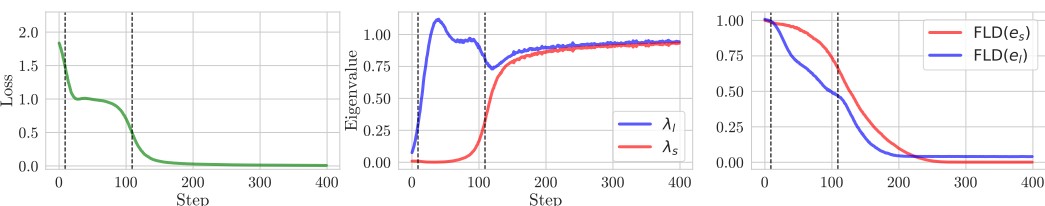

Figure 16: **Effects of extent bias on learning dynamics in non-linear network.** (Left) Stepwise learning curves of Barlow Twins. There are two ($d = 2$) learning steps shown with two black dashed vertical lines (also shown in the other two panels) on empirical result (solid green). (Center) Evolution of eigenvalues $\lambda_j$'s of $C$ during training. At the beginning, the first eigenvalue $\lambda_1$ (blue) increases to 1 and then later the second $\lambda_2$ (red) follows. (Right) Evolution of the feature learning distance FLD($e$) for $e_l$ (blue) and $e_s$ (red). See Definition I.1. We use $m_l = 9$, $m_s = 1$. See Appendix A.1 for more detailed settings.

From Figure 16, we observe FLD($e_l$) drop earlier than FLD($e_s$). Therefore, the phenomenon of $e_l$ being learned before $e_s$ is consistent with the linear case.

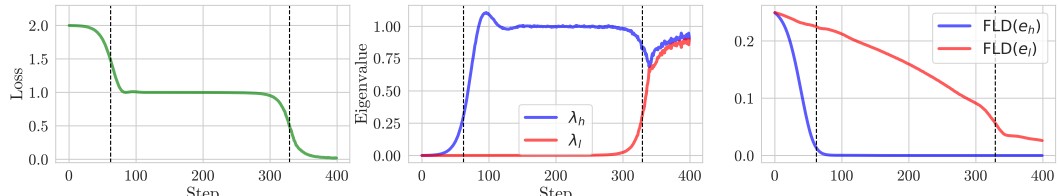

Figure 17: **Amplitude bias effects on learning dynamics in non-linear network.** (Left) Stepwise learning curves of Barlow Twins showing two distinct learning phases with vertical dashed lines marking critical transition points during training. The green line shows empirical loss decreasing in two clear stages. (Center) Evolution of eigenvalues $\lambda_j$ of correlation matrix C during training. The eigenvalue $\lambda_h$ (blue) increases first, followed by the eigenvalue $\lambda_l$ (red), demonstrating amplitude-based learning prioritization. (Right) Evolution of feature learning distance FLD$(e)$ for high-amplitude feature el (blue) and low-amplitude feature es (red), confirming that features with higher amplitude coefficients ($c_h$) are learned before those with lower amplitude ($c_l$), even in non-linear architectures. Note that FLD decreases as the network learns to represent the corresponding feature. We use $c_h = 1, c_l = 0.5$ and a 3-layer network with leakyReLU activations. See the caption of Figure 16. See Appendix A for additional experimental details.

### I.3 Non-linear on amplitude bias

Using Definition I.1, we experiment on Section 5.7. Figure 17 demonstrates amplitude bias effects in non-linear networks. The results show that features with higher amplitude ($c_h$) are learned before those with lower amplitude ($c_l$), consistent with our linear model findings. Specifically, $FLD(e_h)$ decreases earlier than $FLD(e_l)$, mirroring the eigenvalue increase patterns observed in the left and center panels. These results confirm that amplitude bias persists in non-linear architectures, suggesting that amplitude magnitude remains a primary determinant of feature learning priority regardless of network complexity. This provides additional evidence that deep learning models respond more sensitively to amplitude characteristics than frequency properties, even when non-linearities are introduced.

### I.4 Eigenvalues on Shift Augmentation

$$x_{base} = c_a \sin(f_a t + \epsilon_a) + c_b \sin(f_b t + \epsilon_b)$$
$$\epsilon_a, \epsilon_b \overset{\text{i.i.d.}}{\sim} U(-\pi, \pi)$$

$$\Gamma = \mathbb{E}[x_{base} x_{base}^\top]$$
$$\Gamma_{ij} = \mathbb{E}[c_a^2 \sin(f_a i + \epsilon_a) \sin(f_a j + \epsilon_a) + c_a c_b \sin(f_a i + \epsilon_a) \sin(f_b j + \epsilon_b)$$
$$+ c_a c_b \sin(f_b i + \epsilon_b) \sin(f_a j + \epsilon_a) + c_b^2 \sin(f_b i + \epsilon_b) \sin(f_b j + \epsilon_b)]$$

$$\begin{aligned}
\mathbb{E}_{\epsilon_a, \epsilon_b}[\sin(\theta_a + \epsilon_a) \sin(\theta_b + \epsilon_b)] &= \mathbb{E}_{\epsilon_a, \epsilon_b}[\text{Im}(\exp(i(\theta_a + \epsilon_a))) \text{Im}(\exp(i(\theta_b + \epsilon_b)))] \\
&= \mathbb{E}_{\epsilon_a}[\text{Im}(\exp(i(\theta_a + \epsilon_a)))]\mathbb{E}_{\epsilon_b}[\text{Im}(\exp(i(\theta_b + \epsilon_b)))] \\
&= \text{Im}(\mathbb{E}_{\epsilon_a}[\exp(i(\theta_a + \epsilon_a))]) \text{Im}(\mathbb{E}_{\epsilon_b}[\exp(i(\theta_b + \epsilon_b))]) \\
&= \text{Im}(\mathbb{E}_{\epsilon_a}[\exp(i\epsilon_a) \exp(i\theta_a)]) \text{Im}(\mathbb{E}_{\epsilon_b}[\exp(i\epsilon_b) \exp(i\theta_b)]) \\
&= \text{Im}(\varphi(1) \exp(i\theta_a)) \text{Im}(\varphi(1) \exp(i\theta_b))
\end{aligned}$$

We can define $u, d$ as $u = \mu + \alpha, d = \mu - \alpha, \alpha = 2\pi$.

$$\varphi(1) = \frac{\exp(iu) - \exp(id)}{i(u - d)} = \frac{\exp(i\mu)}{\alpha i} \frac{\exp(i\alpha) - \exp(-i\alpha)}{2i} = \frac{\exp(i\mu)}{\alpha i} \sin(\alpha) = 0$$

So,

$$\mathbb{E}_{\epsilon_a, \epsilon_b}[\sin(\theta_a + \epsilon_a) \sin(\theta_b + \epsilon_b)] = 0$$

507 Similar,

$$\mathbb{E}[\sin(\theta_a + \epsilon_a)\sin(\theta_b + \epsilon_a)] = -\frac{1}{2}\mathbb{E}[\cos(\theta_a + \theta_b + 2\epsilon_a) - \cos(\theta_a - \theta_b)]$$

$$= -\frac{1}{2}\mathbb{E}[\cos(\theta_a + \theta_b + 2\epsilon_a)] + \frac{1}{2}\cos(\theta_a - \theta_b)$$

$$= -\frac{1}{2}\int_a^b [\frac{1}{b-a}\cos(\theta_a + \theta_b + 2x)dx] + \frac{1}{2}\cos(\theta_a - \theta_b)$$

$$= -\frac{1}{4}\frac{1}{b-a}[\sin(\theta_a + \theta_b + 2b) - \sin(\theta_a + \theta_b + 2a)] + \frac{1}{2}\cos(\theta_a - \theta_b)$$

$$= -\frac{1}{4}\frac{1}{b-a}[2\cos(\theta_a + \theta_b + a + b)\sin(b-a)] + \frac{1}{2}\cos(\theta_a - \theta_b)$$

508 we assumed $b - a = 2\pi$,

$$\mathbb{E}[\sin(\theta_a + \epsilon_a)\sin(\theta_b + \epsilon_a)] = \frac{1}{2}\cos(\theta_a - \theta_b)$$

509 finally, we get

$$\Gamma_{ij} = \frac{c_a^2}{2}\cos(f_a(i-j)) + \frac{c_b^2}{2}\cos(f_b(i-j))$$

510 is symmetric circulant matrix when $f_a = a\frac{2\pi}{N}, f_b = b\frac{2\pi}{N}$,

$$c_j = \frac{c_a^2}{2}\cos(f_a j) + \frac{c_b^2}{2}\cos(f_b j)$$

$$\Lambda_{\Gamma,k} = \sum_{j=0}^{N-1} c_j \omega^{-kj}$$

$$V_{\Gamma,k} = \frac{1}{\sqrt{N}}\left[1, \omega^k, \omega^{2k}, \ldots, \omega^{(N-1)k}\right]^\top$$

$$\omega = \exp(\frac{2\pi i}{n}) = \cos(\frac{2\pi}{n}) + i\sin(\frac{2\pi}{n})$$

511 This is symmetric, so eigenvalues are real. The eigenvectors can be expressed either in complex form
512 or as pairs of real vectors. Using properties of Discrete Fourier Transform (DFT) matrix on $\Lambda_{\Gamma,k}$,

$$\Lambda_{\Gamma,k} = \begin{cases} 0 \ (k \neq l_a, N - l_a, l_b, N - l_b) \\ \frac{c_a^2}{2} \ (k = l_a \text{ or } k = N - l_a) \\ \frac{c_b^2}{2} \ (k = l_b \text{ or } k = N - l_b) \end{cases}$$

513 Finally, we can derive as:

$$\Lambda_\Gamma = \text{diag}\left(\left[\frac{c_a^2}{2}, \frac{c_a^2}{2}, \frac{c_b^2}{2}, \frac{c_b^2}{2}, \mathbf{0}_{m-2}\right]\right),$$

$$V_\Gamma^{(\leq 4)} = \left[\frac{1}{\sqrt{N}}e_{h,\cos} \ \frac{1}{\sqrt{N}}e_{h,\sin} \ \frac{1}{\sqrt{N}}e_{l,\cos} \ \frac{1}{\sqrt{N}}e_{l,\sin}\right].$$

514 where

$$e_{h,\cos} = c_a \cos(f_a t),$$
$$e_{h,\sin} = c_a \sin(f_a t),$$
$$e_{l,\cos} = c_b \cos(f_b t),$$
$$e_{l,\sin} = c_b \sin(f_b t).$$

