# OpenReview forum: "Stepwise Feature Learning in Self-Supervised Learning"
_NeurIPS.cc/2025/Conference — Submitted to NeurIPS 2025_

### Official Review · Reviewer_F8k6 · 2025-07-03

**Clarity:** 3
**Significance:** 4
**Originality:** 3
**Rating:** 4
**Confidence:** 4

**Summary:**

This paper presents a theoretical framework to explain shortcut learning in self-supervised learning (SSL), specifically through the lens of stepwise feature learning dynamics. The authors propose that during SSL training, models preferentially learn features with higher spatial extent (extent bias) or larger amplitude (amplitude bias), regardless of their semantic relevance. By analyzing the eigendecomposition of the feature cross-correlation matrix in toy Barlow Twins models, the paper demonstrates how features with larger eigenvalues are learned earlier due to the dynamics of gradient flow and weight singular value evolution. These findings are validated in both linear and non-linear (MLP) settings, and further tested on a Colored-MNIST dataset, where models are shown to initially rely on background features before eventually learning object-relevant ones. The work provides theoretical and empirical insights into the biases driving SSL training, with implications for designing future SSL methods that avoid such shortcuts and prioritize generalizable features.

**Questions:**

(1) The organization of the amplitude-bias section could be improved

The transition from frequency to amplitude bias in Section 5 is somewhat abrupt. Although both arise from the same motivation and experimental setup, the narrative jumps between them without first clearly introducing amplitude bias. We suggest restructuring the section to improve clarity.

(2) Interpretability and Practical Implications

While the paper provides strong theoretical insights, its practical implications remain unclear. Could the authors summarize key takeaways for practitioners designing or training SSL models? Additionally, although extent and amplitude bias are well-diagnosed, the paper does not propose mitigation strategies. It would be helpful to discuss how the theoretical framework might inform methods (e.g., regularization or augmentation) to encourage learning of more generalizable features.

(3) Broader Empirical Validation

The analysis primarily relies on toy settings with synthetic data. To strengthen external validity, could the authors include or comment on experiments using more realistic or semi-synthetic datasets? A systematic study across multiple datasets or SSL frameworks (e.g., SimCLR, BYOL) would help verify whether extent and amplitude bias persist in practice and enhance the relevance of the findings.

**Ethical Concerns:**

["NO or VERY MINOR ethics concerns only"]

**Final Justification:**

I initially gave this paper a score of 5 based on the strength of its theoretical contribution. However, after considering the limited empirical evidence in the current version and reading the reviewer discussion, I have adjusted my score to a 4. I still believe the work makes a meaningful and potentially impactful contribution, and I encourage the authors to incorporate the missing empirical validations and practical insights in the revised version to fully realize the paper’s potential.

**Limitations:**

Yes

**Quality:**

3

**Strengths And Weaknesses:**

Strengths:

This paper provides a clear and original theoretical framework that connects shortcut learning in self-supervised learning (SSL) to stepwise feature learning governed by feature extent and amplitude. The use of eigendecomposition and gradient dynamics to explain learning order is both elegant and insightful. The work is well-structured, and its conclusions are consistently validated across linear models, MLPs, and a synthetic Colored-MNIST dataset. These findings offer meaningful insights into the mechanisms underlying spurious feature learning in SSL.

Weaknesses:

The primary limitation lies in the reliance on toy models and synthetic datasets, which may reduce the applicability of the conclusions to real-world SSL settings. The practical implications—how to mitigate extent or amplitude bias—are not fully explored. Additionally, the theoretical framework assumes idealized conditions (e.g., feature independence, aligned initialization) that may not hold in complex, modern architectures. More empirical evidence on realistic datasets would strengthen the impact.

---

> ### Author Rebuttal · Authors · 2025-07-29
>
> Thank you for your thorough and insightful review of our work. We greatly appreciate your time and effort in assessing our paper and providing constructive feedback.
>
> > [W1] The primary limitation lies in the reliance on toy models and synthetic datasets, which may reduce the applicability of the conclusions to real-world SSL settings. The practical implications—how to mitigate extent or amplitude bias—are not fully explored. Additionally, the theoretical framework assumes idealized conditions (e.g., feature independence, aligned initialization) that may not hold in complex, modern architectures. More empirical evidence on realistic datasets would strengthen the impact.
>
> > [Q3] Broader Empirical Validation
> The analysis primarily relies on toy settings with synthetic data. To strengthen external validity, could the authors include or comment on experiments using more realistic or semi-synthetic datasets? A systematic study across multiple datasets or SSL frameworks (e.g., SimCLR, BYOL) would help verify whether extent and amplitude bias persist in practice and enhance the relevance of the findings.
>
> [A1, QA3] Our theoretical findings demonstrate **strong generalizability to realistic settings**, as evidenced by validation across multiple SSL frameworks and practical datasets.
> - On the "Idealized Conditions"
>     - **Feature Independence**: It is a **widely used** assumption.
>     - **Aligned Initialization** : Aligned initialization **is generic and is not idealized conditions**, which shows the robustness of the findings. To follow the aligned trajectory, all we need is to start from a small initialization, proven in Sec. C of Simon et al.
> Our experiments (Sec. 4.5, Fig. 10) demonstrate that models initialized with small random weights rapidly achieve alignment in early training stages. The subspace alignment metric quickly approaches 1, confirming that the aligned structure emerges naturally from small random initialization.
>
> - On the Applicability to Realistic Settings: **The uncovered principles are broadly generalizable**. Our choice of Barlow Twins was motivated by its representative loss structure; as noted by [1], other prominent SSL methods (e.g., InfoNCE, SimCLR and VICReg) also share the same quadratic form $\text{Tr}[WAW^\top]+\text{Tr}[(WBW^\top)^2]$, making it a suitable canonical model for analysis.
> Our empirical results, including new experiments, validate that our findings are not artifacts of this simplified setup but are indeed a **more fundamental phenomenon**.
>     - [**Beyond Barlow Twins**]: The phenomenon is not limited to Barlow Twins. In additional experiments, we observed the identical stepwise learning and the extent bias in a 1-layer **SimCLR** and **VICReg** model. Furthermore, our experiments on the Colored-MNIST dataset using SimCLR and VICReg reproduced the same results, confirming the behavior holds across different SSL frameworks.
>     - [**Beyond Single-Layer and Linear Encoder**]: The phenomenon is not limited to shallow or linear models. We have verified that the same learning priorities manifest in **deep linear networks** (with depth = 4). More importantly, as shown in Sec. 4.7, 4.8 and 5.7 of our manuscript, this behavior persists in **non-linear networks (e.g., ResNet-18, ViT-B/16)**. The fundamental learning priority—which features are learned first—remains consistent. This trend becomes even more apparent when evaluating with linear probing accuracy rather than loss.
>     - [**Beyond Synthetic Datasets**]: **Colored-MNIST** dataset is not "highly simplified" but is a standard benchmark for evaluating model robustness against spurious correlations. We also conduct experiments on **Waterbirds** dataset, made of Caltech-UCSD Birds-200-2011 and Places dataset. Our findings on these datasets further strengthen the claims.
>
> The consistent results across different SSL methods (Barlow Twins, SimCLR, VICReg), network architectures (linear, deep linear, non-linear), and datasets (synthetic, Colored-MNIST, Waterbirds) provide strong evidence that extent and amplitude bias are fundamental principles of self-supervised learning. We will incorporate these additional results and clarifications into the revised manuscript.
>
>
> ---
> > [Q1] The organization of the amplitude-bias section could be improved
> The transition from frequency to amplitude bias in Section 5 is somewhat abrupt. Although both arise from the same motivation and experimental setup, the narrative jumps between them without first clearly introducing amplitude bias. We suggest restructuring the section to improve clarity.
>
> [QA1] We will restructure the section to provide a clearer introduction to amplitude bias and a smoother narrative flow that better explains how our frequency investigation leads to the key insight that amplitude, rather than frequency, drives learning priority.
>
> ---
> > [Q2] Interpretability and Practical Implications
> While the paper provides strong theoretical insights, its practical implications remain unclear. Could the authors summarize key takeaways for practitioners designing or training SSL models? Additionally, although extent and amplitude bias are well-diagnosed, the paper does not propose mitigation strategies. It would be helpful to discuss how the theoretical framework might inform methods (e.g., regularization or augmentation) to encourage learning of more generalizable features.
>
> [QA2] We propose some **practical implications** based on our findings:
> - Extent Bias: We propose data augmentation that partially mask or crop the background, or that increase the size of the object. This would directly manipulate the object's "extent bias" to alter its learning priority.
> - Amplitude Bias: We suggest methods to amplify or reduce the amplitude of specific frequency bands where key features are presumed to exist (e.g., for zebra stripes). Alternatively, one could identify high-amplitude bands in a dataset to check for potential shortcuts.
>
> ---
> [1] Ziyin et al., What shapes the loss landscape of self-supervised learning?, (arXiv preprint)

---

> > ### Comment · Reviewer_F8k6 · 2025-08-07
> >
> > Thank you to the authors for their detailed response and for clarifying the broader applicability of their theoretical framework. I appreciate the effort to connect the core mechanism—extent and amplitude bias—to other SSL methods (e.g., SimCLR, VICReg), deeper architectures (e.g., ResNet-18, ViT-B/16), and real-world datasets such as Waterbirds.
> >
> > That said, I find the current empirical support for these broader claims still somewhat limited. While I understand that no new figures can be added during the discussion phase, many of the rebuttal claims—such as the persistence of extent/amplitude bias in non-linear models or natural image datasets—are not yet supported by inspectable results in the current submission. This makes it difficult to fully evaluate the generalizability and practical impact of the proposed findings beyond the linear/toy settings.
> >
> > I also appreciate the clarification that subspace alignment can emerge from small random initialization, but this remains an important modeling assumption. A more systematic analysis of its robustness across architectures or initializations would help substantiate its general relevance.
> >
> > Similarly, while the authors propose promising directions for mitigating these biases (e.g., cropping backgrounds or adjusting frequency amplitudes), these implications remain hypothetical without supporting empirical analysis or ablation studies.
> >
> > I initially gave this paper a 5 based on the strength of its theoretical contribution and the clarity of its core insight. However, after reading the rebuttal and considering the broader reviewer discussion, I have adjusted my score to 4. I still view the paper as a meaningful contribution with strong theoretical foundations, and I encourage the authors to incorporate the missing empirical evidence and expanded discussions into the revised version. Doing so would significantly strengthen the overall impact and practical relevance of the work.

---

### Official Review · Reviewer_1XYi · 2025-07-04

**Clarity:** 4
**Significance:** 2
**Originality:** 3
**Rating:** 4
**Confidence:** 4

**Summary:**

The authors extend the theoretical analysis by Simon et al. to demonstrate the feature learning biases that exist in self-supervised learning setups, e.g. BarlowTwins. The work is well-positioned with references to empirically-observed feature learning issues in supervised learning, and introduces well-constructed theoretical settings to study equivalent pitfalls in self-supervised learning. Moreover, the authors verify their theoretical analysis with empirical results in simplistic linear network settings, as well as non-linear network settings. Overall, this work presents a neat theoretical analysis to study commonly observed pitfalls in feature learning, which might be interesting to the community.

**Questions:**

1. While the data model in Section 4.1 and 5.1 are interesting and makes the theoretical analysis easy, how general do you think they are for realistic datasets? Can you provide an example where this can be applied?
2. The notations of $f_h$ and $f_l$ are confusing because it makes the reader think that they correspond to high and low frequency respectively. Is there a specific reason to use this notation? Can you change it to $f_{ha}$ and $f_{la}$ to indicate high and low amplitude, respectively?
3. In the derivation in Appendix B.2, is there a missing $f$ in the denominator (after integrating $cos(2fx)$)? I don't think it changes the conclusion, but curious if it's a typo or something that I missed.
4. The discussions section talks about the absence of frequency shortcut in BarlowTwins models, as opposed to supervised learning models. Can you comment more on the reason for this? Is it possible to talk about this as a pro over supervised learning (or maybe cross-entropy type losses)? Could this be related to work from Balestriero & Lecun [3]?

- [3] Balestriero et al. How Learning by Reconstruction Produces Uninformative Features For Perception. ICML 2024.

**Ethical Concerns:**

["NO or VERY MINOR ethics concerns only"]

**Final Justification:**

Based on my discussion with the authors, I feel their responses have addressed most of my comments. However, I feel the submission has undergone significant modifications, some of which are empirical and hard to evaluate in the NeurIPS rebuttal format, and would benefit from another round of reviews. In its current state, i.e. with the modifications duly incorporated, this work is a significant contribution and will have a large impact on the SSL community.

I am happy to defer to the AC's recommendation on how to take the recent (significant) modifications into account for making a final decision on the paper.

**Limitations:**

Yes, addressed.

**Quality:**

3

**Strengths And Weaknesses:**

Strengths:
1. The paper is very well-written and easy to follow. The claims are described clearly and supported with proper theoretical analysis and empirical results.
2. The theoretical problem setup is nicely chosen and explained. While the data model might be restrictive for practical purposes, it helps the reader appreciate the theoretical contributions and gain an intuitive understanding of the underlying phenomenon.
3. The empirical results in non-linear setting are a good addition, and the colored-MNIST experiment is a fantastic example of a controlled experiment. Unfortunately, such experiments are rare in the deep learning literature, so kudos to the authors for crafting such an experiment and demonstrating their core claim.

Weaknesses:
1. I think a core issue with this work (that limits me from rating it higher) is its applicability. The current results are limited, both in terms of experimental evaluation in larger networks and complex datasets as well as the theoretical setup. Specifically, the authors extended the dynamics studied by Simon et al. for the BarlowTwins loss with a high redundancy reduction coefficient. In practice, the redundancy reduction coefficient is significantly smaller, and the resulting dynamics are distinct. Recent work [1] has studied these distinctions, which might be of interest to the authors. Therefore, I would recommend the authors to extend their analysis following the results of Ghosh et al. (see Theorems 3.2 & B.4). I believe this work would greatly benefit from comparing the feature learning behaviour studied by the authors in their setup for the high vs low redundancy reduction coefficient cases.
2. While the data model presented here is better for theoretical advances, I feel it is restrictive to study the feature learning properties in realistic datasets. As a result, I am unsure how useful the results of this study will be to advance practical SSL pipelines. For instance, can the authors provide recommendations on how to improve the feature learning properties by changing the data augmentation strategy or with some dataset preprocessing?
3. While the non-linear network experiments demonstrate that the theoretical results from linear networks hold for non-linear networks as well (which is very interesting and great to see), the eigenvalue dynamics seem to be different. Unfortunately, the authors do not comment on this discrepancy in the current version of the work. Also, I would recommend the authors to consider moving these results to the main text instead of the appendix. Furthermore, I also feel this is a great opportunity to talk about the effect of non-linearity on feature learning in intermediate layers of the network and study the distinctions in learning dynamics of features vs embeddings in BarlowTwins. Recent work [2] has made some progress in this direction, which might be of interest to the authors.

Overall, I feel this work is exciting (both personally and overall for the field) and presents some interesting insights for the self-supervised learning community. However, the discrepancy from practical settings limits the utility of the results presented here. If the authors can resolve some of these discrepancies, I will be happy to reconsider my score.

- [1] Ghosh et al. Harnessing small projectors and multiple views for efficient vision pretraining. NeurIPS 2024.
- [2] Xue et al. Investigating the Benefits of Projection Head for Representation Learning. ICLR 2024.

---

> ### Author Rebuttal · Authors · 2025-07-29
>
> Thank you for your thorough and insightful review, and feedback.
>
> > [W1] Core issue is its applicability. The current results are limited, both in terms of experimental evaluation in larger networks and complex datasets as well as the theoretical setup. The authors extended the dynamics studied by Simon et al. for the BarlowTwins loss with a high redundancy reduction coefficient. In practice, the redundancy reduction coefficient is significantly smaller, and the resulting dynamics are distinct. Recent work [1] has studied these distinctions, which might be of interest to the authors. Therefore, I would recommend the authors to extend their analysis following the results of Ghosh et al.
>
> [A1] We confirmed that the **same phenomenon (e.g., extent bias)** occurs even in larger networks, complex dataset, and low redundancy reduction coefficient.
> - Large network architecture : We validated our findings using **ResNet-18, ViT-B/16**, standard deep architectures, rather than analysis to 1-layer linear networks.
> - Complex dataset validation :
>     - **C-MNIST** : Fig 2 demonstrates the predicted extent bias behavior across different object pixel ratios (0.05, 0.10, 0.15). Models exhibit rapid initial learning of background features (achieving 70% accuracy plateau), followed by gradual object feature learning, with plateau duration inversely related to object pixel ratio.
>     - **Waterbirds** [4] : We additionaly experiments waterbirds dataset, which is dataset made of Caltech-UCSD Birds-200-2011 and Places dataset. We downsampling 256x256 image size to 128x128. Following the same experimental design on C-MNIST, we set bird-to-image pixel ratio to 0.1 and observed extent bias effects. The background features, having larger extent bias than bird features, showed plateau periods in spurious correlation prediction accuracy (setted by 59.5%), confirming that the phenomenon generalizes beyond synthetic datasets to realistic natural image benchmarks widely used in the spurious correlation literature.
>
> - Low redundancy reduction coefficient $\lambda$ : We confirmed that the **same phenomenon (e.g., extent bias)** occurs even in a single-layer linear network with a **low redundancy reduction coefficient $\lambda$, such as 0.005**. Actually, a typical $\lambda = 0.005$ is used in Fig 2 (Colored-MNIST, ResNet-18). Herefore, while the precise learning dynamics might differ from our simplified theoretical model, our empirical evidence suggests the central conclusion holds: features corresponding to higher eigenvalues are learned preferentially, regardless of the specific dynamics.
>
> ---
> > [W2] It is restrictive to study the feature learning properties in realistic datasets. As a result, I am unsure how useful the results of this study will be to advance practical SSL pipelines. For instance, can the authors provide recommendations on how to improve the feature learning properties by changing the data augmentation strategy or with some dataset preprocessing?
>
> [A2] For **practical SSL pipeline**, we will expand the discussion section to suggest a potential application of our research. For instance, based on our findings on extent bias, applying preprocessing techniques such as cropping the outer background of an object or masking background—which effectively increases the object's relative extent bias—could encourage the model to easily learn more object-centric features.
>
> ---
> > [W3] The non-linear network eigenvalue dynamics seem to discrepancy. Recommand moving these results to the main text instead of the appendix. The effect of non-linearity on feature learning in intermediate layers of the network, and the distinctions in learning dynamics of features vs embeddings in BarlowTwins.
>
> [A3] The **fundamental learning priority is maintained**, where features corresponding to larger eigenvalues are learned first.
>
> [**Effect of non-linearity on feature learning in intermediate layers of the network**]
> Eigenvalue overshoot phenomenon is observed in the non-linear MLP models. Non-linear activations cause this discrepancy because non-linear activation in the intermediate layers breaks the symmetry between layers [1].
>
> [**Distinctions in learning dynamics of features vs embeddings in BarlowTwins**]
> To understand the differences between embeddings and features, we examine the **kernelized perspective**. Given input data, $X=[x_1, \dots, x_d]^\top \in\mathbb{R}^{n \times m}$ and kernel matrix $\tilde K = [X;X'][X^\top X'^\top] \in \mathbb{R}^{2n \times 2n}$, the eigenpair $(\lambda, g)$ of $\Gamma$ where $\Gamma g = \gamma g$ has a corresponding kernelized eigenvector $b = \tilde K^{-1/2}[X;X']g$. The kernel matrix $K_\Gamma \equiv \tilde K^{-1/2} Z \tilde K^{-1/2}$ where $Z=[X;X']\Gamma[X^\top X'^\top]$ preserves the duality relationship $(\gamma, g) \Leftrightarrow (\gamma, b)$ as proven in Sec B.2 in Simon et al. (2023).
> This ensures learning the kernel eigenvector $b$ with $\gamma$ corresponds to learning the feature eigenvector $g$. However, the specific features that achieve higher eigenvalues depend on both architectural inductive biases and augmentation strategies through their invariance properties.
> Architecture can be assumed as a kernel $K$ that acts as an implicit feature transformation $\phi(x)$, $K(x, x') = \phi(x)^\top\phi(x')$. Features more aligned with the architectural inductive bias - those that remain relatively invariant under the implicit transformation $\phi$ - exhibit larger eigenvalues and be prioritized during learning.
> - Features ($x$): Smooth, monotonic eigenvalue increase following theoretical predictions. Direct eigenvalue-based priority (extent/amplitude bias).
> - Embeddings ($\phi(x)$): Eigenvalue overshoot phenomenon due to non-linear activations. Same priority maintained but influenced by architectural/augmentational inductive biases.
>
> The fundamental learning prioritization mechanism remains robust, features with larger eigenvalues are learned first in both cases, but non-linear transformations introduce complex intermediate dynamics while preserving the core learning order.
>
> ---
> Following your valuable suggestion, we will move the non-linear network experiments from the appendix to the main text to better highlight the robustness of our findings.
>
> ---
> > [Q1] While the data model in Sec 4.1 and 5.1 are interesting and makes the theoretical analysis easy, how general do you think they are for realistic datasets? Can you provide an example where this can be applied?
>
> [QA1] We believe our findings are quite generalizable to realistic datasets. For instance, in Table 13 of paper [2], SimCLR's test accuracy drops to as low as 19.2%. This demonstrates a real-world scenario where a model prioritizes background cues, not objects, which aligns with our analysis.
>
> ---
> > [Q2] The notations of $f_h$ and $f_l$ are confusing. Can you change it to $f_{ha}$ and $f_{la}$ to indicate high and low amplitude?
>
> [QA2] I acknowledge that $f_h$ and $f_l$ notation could be misleading. Following your suggestion, this will be corrected to $f_{ha}$ and $f_{la}$ ​ to clearly indicate high and low amplitude.
>
> ---
> > [Q3] In the derivation in Appendix B.2, is there a missing f in the denominator (after integrating $cos(2fx)$)? I don't think it changes the conclusion, but curious if it's a typo or something that I missed.
>
> [QA3] Yes, there is a missing factor of $f$ in the denominator. The integration should yield $\sin(2fx)/2f$​, not $\sin(2fx)/2$​. Though as you noted, it doesn't affect the final conclusion since the $\sin⁡(2fm)$ term still vanishes in the limit or due to the periodicity argument. This will be corrected.
>
> ---
> > [Q4] The discussions section talks about the absence of frequency shortcut in BarlowTwins models, as opposed to supervised learning models. Is it possible to talk about this as a pro over supervised learning (or maybe cross-entropy type losses)? Could this be related to work from Balestriero & Lecun?
>
> [QA4] Neural network learning dynamics **fundamentally differ across different tasks**, reflecting the distinct objectives and constraints inherent to each learning paradigm:
> - **[Supervised learning (Regression tasks)]**: models are known to learn low-frequency components first (frequency bias or spectral bias).
> - **[Supervised learning (Classification task)]**: models often develop a "frequency shortcut," as demonstrated in the paper "What do neural networks learn in image classification?" [3]. This means they prioritize learning distinctive frequency patterns that make it easy to distinguish between classes.
> - **[Self-supervised learning]**: Instead of learning based on frequency, SSL models prioritize features with high amplitude. The key distinction is that classification learns based on which frequencies are most distinguishable for classification, while SSL learns based on which features have the highest amplitude.
>
> Pros over supervised learning (Classification specific) : The model can be **less biased towards specific frequency patterns**. For example, consider a person wearing a shirt with a zebra pattern. A classification model might use the high-frequency stripes as a shortcut and misclassify the image as a "zebra." However, an SSL model learns based on amplitude. If the the person's outline had a higher amplitude in the training data than the feature for the zebra pattern, the model would prioritize the person's shape, potentially reducing the chance of such a misclassification.
>
> ---
> [1] Nam et al., Position: Solve Layerwise Linear Models First to Understand Neural Dynamical Phenomena (Neural Collapse, Emergence, Lazy/Rich Regime, and Grokking), (ICML 2025)
>
> [2] Qiang et al., On the Out-of-Distribution Generalization of Self-Supervised Learning, (ICML 2025)
>
> [3] Wang et al., What do neural networks learn in image classification?, (ICCV 2023)
>
> [4] Sagawa et al., Distributionally Robust Neural Networks for Group Shifts: On the Importance of Regularization for Worst-Case Generalization, (ICLR 2020)

---

> > ### Comment · Reviewer_1XYi · 2025-08-03
> > **Official comment by Reviewer 1XYi**
> >
> > I would like to thank the authors for responding to my comments and questions. I believe the proposed clarifications and typo corrections will help improve the overall quality and clarity of the work.
> >
> > Some final thoughts:
> > 1. The new empirical results seem to be huge modification over the original submission. Unfortunately, owing to NeurIPS regulations, it is hard for me to judge the results and I have to take the authors' word for it (as opposed to look at the plots and see the similarities). It is a shame, but also indicates that the authors have significantly strengthened the empirical section in the period since the work was submitted. While this is commendable, I wonder if the work will benefit from undergoing another round of review at a different conference where the updated version of the work (with the new results) are more fairly evaluated. I believe that the new empirical results definitely add a big boost to the overall quality of the submission.
> >
> > 2. While I appreciate the empirical support provided by authors in showing that their results extend to low $\lambda$ settings, I would expect an update to the theoretical section to encompass this setting. Given that a core contribution of this work is its theoretical results, I believe extending the theoretical section with the dynamics for low $\lambda$ setting would help preserve the core contributions of the paper. Same goes for the effect of non-linearity. The current description seems more of an intuitive explanation, and needs more rigor.
> >
> > 3. I like the authors' summarization of the pros of BarlowTwins loss over supervised losses. However, the example seems fairly conjectural and needs targeted experiments. Same goes for the authors' response to Q1. It's hard to follow how the authors relate the results from Qiang et al. (I assumed you meant to refer to Table 12?) to their analysis directly. I feel further experiments and analysis are required to ensure that the drop in SimCLR (and BarlowTwins) performance is indeed due to prioritization of background cues.
> >
> > Overall, I like the spirit of the work as it seems to be striking a balance between theory and empirical results. I also think this line of work is very interesting and has great potential (hence my rating). However, the new results are harder to evaluate and the theoretical results/justifications do not adequately address my comments.

---

> > > ### Author Response · Authors · 2025-08-06
> > > **Response to 1XYi [A1, A3]**
> > >
> > > Thank you for your comprehensive review and valuable feedback. We appreciate your recognition of our work's potential and the balance between theory and empirics.
> > >
> > > ---
> > > >The new empirical results seem to be huge modification over the original submission. Unfortunately, owing to NeurIPS regulations, it is hard for me to judge the results and I have to take the authors' word for it (as opposed to look at the plots and see the similarities). It is a shame, but also indicates that the authors have significantly strengthened the empirical section in the period since the work was submitted. While this is commendable, I wonder if the work will benefit from undergoing another round of review at a different conference where the updated version of the work (with the new results) are more fairly evaluated. I believe that the new empirical results definitely add a big boost to the overall quality of the submission.
> > >
> > > [A1] We appreciate your acknowledgment of our empirical evaluation efforts. We understand your concern about evaluating new experiments within the rebuttal format constraints.
> > > However, we respectfully note that our core theoretical contributions - **the formalization of extent/amplitude bias and their role in SSL shortcut learning** - stand independently and are already well-supported by the experiments in the main paper. The additional architecture experiments, while valuable extensions, **represent supplementary** validation rather than fundamental requirements for our theoretical framework.
> > > We believe the current work already makes significant contributions to understanding SSL dynamics, as evidenced by our theoretical analysis now extended to kernelized settings and various augmentations and the consistent experimental validation across multiple settings (general Barlow Twins loss, multiple architectures, and data modalities). We are committed to including all additional results in the camera-ready version to further strengthen the paper's impact.
> > >
> > > ---
> > > >I like the authors' summarization of the pros of BarlowTwins loss over supervised losses. However, the example seems fairly conjectural and needs targeted experiments. Same goes for the authors' response to Q1. It's hard to follow how the authors relate the results from Qiang et al. (I assumed you meant to refer to Table 12?) to their analysis directly. I feel further experiments and analysis are required to ensure that the drop in SimCLR (and BarlowTwins) performance is indeed due to prioritization of background cues.
> > >
> > > [A3] Regarding BarlowTwins advantages, we should clarify: our work identifies that SSL methods (including BarlowTwins) exhibit **stepwise feature learning based on eigenvalue ordering**, where extent bias and amplitude bias. This theoretical insight reveals both strengths and vulnerabilities of SSL - not claiming superiority over supervised learning, but explaining **when and why** SSL methods fail.
> > >
> > > Regarding the Qiang et al. (Table 12), we cited their work because we cannot provide plots due to the Neurips policy. Their table 12 shows SimCLR's performance drop on Waterbirds, which directly supports our extent bias theory. Our framework provides the theoretical explanation for this empirical observation: SSL methods prioritize spatially larger backgrounds (extent bias, higher eigenvalues) over smaller objects (extent bias, lower eigenvalues), leading to failure when backgrounds are spuriously correlated.
> > >
> > > We agree that a targeted experimental comparison between supervised and SSL methods would be a valuable, but separate research direction. Our primary contribution in this work is to provide the foundational theoretical framework that explains *why* such empirical observations in the literature occur.

---

> > > ### Author Response · Authors · 2025-08-06
> > > **Response to 1XYi [A2] Redundancy reduction term in BT.**
> > >
> > > [A2] Redundancy reduction term in BT.
> > >
> > > **Even if $λ \neq 1$, the principle "features with larger eigenvalues are learned earlier" keeps hold, but dynamics differs depending on $λ$.**
> > > First we consider general Barlow Twins loss:
> > > $$L\_λ=\sum\_{i}([W\Gamma W^\top]\_{ii}-1)^2+λ\sum\_{i\neq j}[W\Gamma W^\top]\_{ij}^2=(1-λ)L\_0+λL\_1.$$
> > > Thus it exhibits a mixed dynamics between the $L_0$ and $L_1$. Therefore, we first consider the dynamics of $L_0$ and $L_1$:
> > > - $$L_1=\\|W\Gamma W^\top -I_d\\|_F^2=\sum_j(s_j^2γ_j-1)^2,$$
> > > - $$\frac{dL_1}{ds_k}=4s_kγ_k(\sum\_j{\color{red}{\delta\_{kj}}}s_j^2γ_j-1),$$
> > > - $$L_0=\sum\_{i}([W\Gamma W^\top]\_{ii}-1)^2=\sum\_{i}(e^{(i)\top} W\Gamma W^\top e^{(i)}-1)^2=\sum\_{i}(u^{(i)\top}S\Lambda\_\Gamma S^\top u^{(i)}-1)^2=\sum\_{i}(\sum_j u_j^{(i)2} s_j^2γ_j-1)^2,$$
> > > - $$\frac{dL_0}{ds_k}=\sum\_{i}2(\sum_j u_j^{(i)2}s_j^2γ_j-1)u_k^{(i)2} 2s_k γ_k=4s_kγ_k\sum\_{i}(\sum_j {\color{red}u_j^{(i)2}}s_j^2γ_j-1)u_k^{(i)2}.$$
> > >
> > > If $u_j^{(i)2}=\delta_{ij}$ ($U=I_d$), then $L_0=L_1$.
> > > If not, $\sum_j u_j^{(i)2}=1$ and each $u_j^{(i)2}$ acts as an averaging weight $1/d$.
> > >
> > > We now investigate the dynamics of $s_k$’s:
> > > - Initially, all singular values grow with exponential dynamics as $s_j\approx 0$ and $\delta\_{kj}=0$ for $j\neq k$. $$\dot s_k=-\frac{dL_0}{ds_k}=-4s_k γ_k\sum\_{i}(\sum_j u_j^{(i)2} s_j^2γ_j-1)u_k^{(i)2}=4s_k γ_k+O(s_k^3)$$
> > > $$\dot s_k=-\frac{dL_1}{ds_k}=-4s_kγ_k(\sum\_j\delta\_{kj}s_j^2γ_j-1)=4s_k γ_k+O(s_k^3)$$
> > >
> > > - After a few steps, the first singular value $s_1$ increases (since $γ_1$ is the largest) and then,
> > > $$\dot s\_1=-\frac{dL\_0}{ds\_1}=-4s\_1 γ\_1\sum\_{i}(\sum\_j u\_j^{(i)2} s\_j^2γ\_j-1)u\_1^{(i)2}=4s\_1 γ\_1\left(1-\sum_j (\sum_i u_j^{(i)2} u_1^{(i)2}) s_j^2 γ_j\right)$$
> > > $$=4s\_1 γ\_1\left(1-\sum_i u_1^{(i)4} s_1^2 γ_1\right)+O\left(\max_{j>1}s_j^2\right) \approx 4s\_1 γ\_1\left(1-\frac{1}{d}s\_1^2 γ\_1\right) $$
> > > $$\dot s\_1=-\frac{dL\_1}{ds\_1}=-4s\_1γ\_1(\sum\_j\delta\_{1j}s\_j^2γ\_j-1) =4s\_1 γ\_1 \left(1-s\_1^2 γ\_1\right)$$
> > >     - [$λ=0$] $s_1$ saturates as $s_1^2γ_1$ reaches $d$.
> > >     - [$0<λ<1$] $s_1$ saturates as $s_1^2γ_1$ reaches the harmonic mean $\frac{1}{λ + (1-λ)\frac{1}{d}}$ of 1 and $d$.
> > >     - [$λ=1$] $s_1$ saturates as $s_1^2γ_1$ reaches 1.
> > >
> > > - After the first loss drops where $s_k$'s are still small except for $s_1$
> > > $$\dot s_k=-\frac{dL_0}{ds_k}=-4s_k γ_k\sum\_{i}(\sum_j u_j^{(i)2} s_j^2γ_j-1)u_k^{(i)2}=O(s_k)$$
> > > $$\dot s_k=-\frac{dL_1}{ds_k}=-4s_kγ_k(\sum\_j\delta\_{kj}s_j^2γ_j-1)=4s_k γ_k(1-s\_k^2 γ\_k)$$
> > >     - [$λ=0$] $\dot s_k$ becomes nearly zero, effectively stopping the growth of other singular values. Each $s_k$ ($k \neq 1$) stays near zero.
> > >     - [$0<λ<1$] First, $s_k$ exponentially grows with the following dynamics:
> > > $$\dot s_k = -4s_k γ_k\sum\_{i}(\sum_j u_j^{(i)2} s_j^2γ_j-1)u_k^{(i)2}=4s_k γ_k\left(1-\sum\_{i}\sum_{j\neq k} u_j^{(i)2}u_k^{(i)2} s_j^2γ_j\right) + O(s_k^3)$$
> > > with the growth rate smaller than that of $λ=1$:
> > > $$4 γ_k\left(1-\sum\_{i}\sum_{j\neq k} u_j^{(i)2}u_k^{(i)2}s_j^2γ_j\right)<4γ_k.$$
> > > Then, each $s_k$ follows sigmoidal dynamics and saturates near some value larger than $\sqrt{1/γ_k}$.
> > >  But after saturation $s_k$ decreases again since the dynamics of $s_k$ is coupled with other singular values. As other singular value increases, the sum $\sum\_{i}\sum_{j\neq k} u_j^{(i)2}u_k^{(i)2}s_j^2γ_j$ grows and exceeds 1, which causes $s_k$ to decrease as the exponent becomes negative.
> > > Note that the principle "features with larger eigenvalues are learned earlier" still holds.
> > >     - [$λ=1$] Each $s_k$ exhibits independent sigmoidal dynamics and saturates as $s_k^2γ_k$ reaches 1.
> > >
> > > ||$λ=0$|$0<λ<1$|$λ=1$|
> > > |-|-|-|-|
> > > |loss|$L_0$|$(1-λ)L_0+λ L_1$|$L_1$|
> > > |dynamics of $s_k(t)$| only $s_1$ changes | each $s_k$ affects each other | independent sigmoidal dynamics|
> > > |in the beginning| $s_1$ shows sigmoidal dynamics, increases and saturates near $\sqrt{d/γ_1}$. | $s_1$ shows sigmoidal dynamics, increases and saturates at $\sqrt{\frac{1}{(λ+(1-λ)\frac{1}{d})γ_1}}$. | $s_1$ shows sigmoidal dynamics, increases and saturates at $\sqrt{1/γ_1}$.|
> > > |$\tau_1$|$\sim \frac{-\log(s_1^2(0)γ_1/d)}{8γ_1}$| $\sim\frac{-\log\left(s_1^2(0)γ_1(λ+(1-λ)\frac{1}{d})\right)}{8γ_1}$|$\sim \frac{-\log(s_1^2(0)γ_1)}{8γ_1}$|
> > > |after the first drop | $\dot s_k\approx 0$ and $s_k$ stays near zero for $k=2,3,\cdots$. | $s_k$ increases sigmoidally and saturates near some value larger than $\sqrt{1/γ_k}$ (and decreases slowly when the next singular value $s_{k+1}$ increases) | $s_k$ shows independent sigmoidal dynamics, increases and saturates at $\sqrt{1/γ_k}$, starting from smaller $k=2,3,\cdots$.|
> > > |$\tau_k$|$\infty$|relatively later than the $λ=1$ case.|$\propto \frac{1}{γ_k}$|
> > >
> > > **On non-linear NNs**:
> > > Fig.14-15 show that non-linear NNs with $λ=1$ exhibit a **coupled singular value dynamics similar to that of linear encoder with $0<λ<1$**. The non-linearlity appears to break the symmetry across layers, resulting in coupled dynamics between singular values that drive (Nam et al. 2025).

---

> > > > ### Comment · Reviewer_1XYi · 2025-08-07
> > > > **Response to authors**
> > > >
> > > > I would like to thank the authors for their hard work in providing clarifications to my concerns. Please find my responses below:
> > > >
> > > > 1. **New empirical results on larger networks and complex datasets:** I agree with the authors that the core contribution of their work is establishing the extent/amplitude bias and their role in SSL shortcut learning. However, I respectfully disagree that this phenomenon is well-supported empirically by the experiments in their original submission. Note that my initial concern was the applicability of the presented data model in realistic datasets. To that end, these new results provide key empirical evidence that elevates the applicability of the work and significantly improves its impact. Therefore, I feel that evaluating these new results is pivotal to rating this work appropriately. I will discuss this situation with other reviewers and the AC before considering these results for my rating.
> > > >
> > > > 2. **Supporting results from Qiang et al.**: I thank the authors for clarifying that they were indeed referring to Table 12 and not Table 13. I re-read the paper from Qiang et al. and can see the connection that the authors are trying to draw from their claims. However, I feel further experiments are required to confirm this connection. For instance, given the authors' theory, features corresponding to the larger eigenvalues are learned earlier in training. If these eigenvalues indeed correspond to the background, could using the feature space learned earlier in training to "clean" the feature space later in training be useful in eliminating this shortcut learning in SSL? In other words, can you adapt the feature space for the network at the end of SSL training to get rid of the subspace that overlaps with the feature space learned early in training and demonstrate that the background bias is removed or reduced? I would consider such an experiment to be more concrete evidence of the authors' claim, especially when they are referring to results from prior literature.
> > > >
> > > > 3. **Role of redundancy reduction term:** Looking at the derivation, it seems that the authors have stumbled upon results similar to that presented by Ghosh et al. While this is great, I feel this is slightly tangential to my comment. I was hoping the authors would connect this result to their extent/amplitude bias phenomenon. From what I can see, it seems that the bias is exacerbated in the $\lambda=0$ case (with only the top eigenvalue being learned) and it is slightly better when $0 < \lambda < 1$ case but still worse than the $\lambda=1$ case. Is that a correct interpretation? I think this is indeed a great addition to the paper and would recommend adding this Table (along with a row for extent/amplitude bias) to the main text (you can have the complete derivation in the appendix and simply provide a citation to prior literature in the main text).
> > > >
> > > > Overall, I think the new results and theoretical contribution are a great addition to the paper. Both of them taken together address my concerns. I will discuss with the other reviewers and AC on how to take the new empirical evidence into consideration, but I would like to maintain my positive rating for this paper. Finally, I would like to congratulate the authors on their great work.

---

> > > > > ### Author Response · Authors · 2025-08-09
> > > > >
> > > > > Thank you for your thorough review and for maintaining a positive rating for our paper. We sincerely appreciate your insightful feedback and constructive suggestions, which have helped us significantly strengthen our work. Below, we outline our responses to your comments.
> > > > >
> > > > > ---
> > > > >
> > > > > > The "Feature Cleaning" Experiment
> > > > >
> > > > > Your suggestion to use the feature space learned early in training (hypothesized to be background features) to "clean" the final feature space is a brilliant and insightful idea. This would provide a **direct causal test** of our theory.
> > > > >
> > > > > Given that this is a non-trivial experiment requiring new methodological design, it was not feasible to complete within the remaining rebuttal period. However, we believe the idea has significant merit. We will experiment with the "feature cleaning" method after rebuttal period.
> > > > >
> > > > > ---
> > > > >
> > > > > > The Role of the Redundancy Reduction Term $λ$
> > > > >
> > > > > Your interpretation of how the redundancy reduction coefficient $λ$ affects the extent/amplitude bias is **correct**.
> > > > >
> > > > > $λ=0$ (Worst Case): Without any penalty on redundancy, the model learns only the single most dominant shortcut feature (the one with the largest eigenvalue) and collapses, exacerbating the bias to its maximum.
> > > > >
> > > > > $0 < λ < 1$ (Practical Case): The bias is still strong. The dynamics are coupled, and the top feature dominates for a prolonged period before other features are slowly learned.
> > > > >
> > > > > $λ=1$ (Idealized Case): The strict orthogonality encourages all features to be learned independently and sequentially, which mitigates the dominance of a single shortcut feature most effectively compared to the other cases.
> > > > >
> > > > > As you suggested, this is an excellent point that deepens our analysis. We will add the summary table comparing the dynamics under different $λ$ values to the main text of the paper, with the full derivation provided in the appendix.
> > > > >
> > > > > We believe these revisions will fully address your remaining concerns. Thank you once again for your constructive engagement and for helping us improve our work.

---

### Official Review · Reviewer_KYPU · 2025-07-05

**Clarity:** 3
**Significance:** 3
**Originality:** 2
**Rating:** 4
**Confidence:** 4

**Summary:**

The work studies shortcut learning in SSL, in relation to the step-wise learning dynamics of a linear Barlow Twins model, under strong orthogonality constraints.

The authors consider two models of data features inducing shortcut learning: those with large spatial extent, and those with large amplitude. Both kinds of features correspond to eigendirections with large eigenvalues in the data covariance matrix, which dominate learning under a gradient-flow model. The resulting theory provides a formal model of extent and amplitude bias, which is matched empirically in toy settings, as well as with non-linear models (MLPs) and ColoredMNIST.

**Questions:**

1. When Barlow Twins (or other equivalent losses) are kernelized, it is possible to study the bias arising from model architectures (which essentially encodes which eigendirections are preferred). Could the authors relate their findings to Cabannes et al. (2023), Proposition 4 (and related discussions)?

2. Again in the kernel setting, the interplay between data augmentation and the redundancy term (controlled by $\lambda$) of the Barlow Twins loss has been discussed in the literature (Ghosh et al. 2024). Could Theorem 4.5 be adjusted to account for Barlow Twins under weaker orthogonality constraints?

3. Data augmentation influences the structure of the data covariance matrix. Several covariance operators have been theoretically discussed in the literature (Lin et al., 2024) beyond the isotropic setting, in a similar toy linear regression model (albeit for supervised learning). How does the noise covariance affect feature learning? Particularly, shouldn’t $\Gamma$ also depend on the noise covariance in Equation 5?

4. In the high-dimensional setting, whereby feature covariances exhibit heavy-tailed behaviour (Agrawal et al., 2022), how likely is the assumption that large superpixels correspond to large-eigenvalues to hold in practice, once features are averaged over a large dataset?

**Minor Questions**

5. In section 4.8, what is the standard digit-to-pixel ratio of ColoredMNIST?

6. Can Theorem 4.3 predict how long the test accuracy plateaus for in Figure 2?


**Minor mistakes**

7. The work of Doersch, et al. appears both as citation 10 and 11.

**References**

* The SSL Interplay: Augmentations, Inductive Bias, and Generalization. Cabannes et al. ICML 2023.
* Harnessing small projectors and multiple views for efficient vision pretraining. Ghosh et al. NeurIPS 2024.
* The good, the bad and the ugly sides of data augmentation: An implicit spectral regularization perspective. Lin et al. JMLR 2024.
* alpha-ReQ : Assessing Representation Quality in Self-Supervised Learning by measuring eigenspectrum decay. Agrawal et al. NeurIPS 2022.

**Ethical Concerns:**

["NO or VERY MINOR ethics concerns only"]

**Final Justification:**

After extensive discussion with me and other reviewers, I believe the authors have fully addressed the outstanding theoretical concerns that were raised in the initial reviews.

The authors have produced additional theoretical results, as well as experiments, which broaden the scope of the manuscript in two ways:

1. Extending the proposed formal analysis to non-linear training dynamics.
2. Validating the core theoretical statements empirically on non-linear models.

Importantly, the authors propose in their rebuttal a new experiment establishing the presence of amplitude and extent bias across different architectures, notably ViT models trained on a modified version of the Waterbirds and Places datasets, where shortcut learning can be induced.

While unfortunately reviewers are unable to fairly evaluate said experiment due to limitations with the NeurIPS format, I believe it does greatly improve the significance and relevance of the work. Due to this reason, I am inclined to recommend the authors to include the updated experiment as a major contribution of the next revision of their work, which would then require a new round of reviewing.

**Limitations:**

Yes.

**Paper Formatting Concerns:**

None.

**Quality:**

3

**Strengths And Weaknesses:**

**Strengths**

1. The paper provides a clear theoretical formalization of two settings contributing to shortcut learning. Formally understanding this problem in the context of SSL is novel and very important, for both practitioners as well as theoretically.

2. Results are presented clearly and concisely, and the paper is generally well written and easy to follow.

3. Results are validated on small non-linear models and toy empirical settings.

**Weaknesses**

1. While the linear model is extremely useful for drawing intuitions, the authors fall short of providing a “kernelized” version. The work of Simon et al. (2023), which the present submission builds upon, provides a kernel version of the toy Barlow Twins model, which could have been exploited by the authors.

2. The proposed linear model cannot account for architecture bias, which heavily influences which eigendirections are learned first and the speed of convergence. This has been studied for SSL in the kernel setting.

3. The problem of subspace alignment is heavily influenced by the loss function, as well as optimizer hyperparameters. Specifically, I would have appreciated a discussion of the role of the redundancy term in the Barlow Twins loss (see Questions).

4. The role of the noise covariance (data augmentation) is not discussed (see Questions).

5. Some results (Theorem 4.2 and 4.3) are direct consequences of Simon et al. (2023), and thus constitute a validation of the prior work’s theory rather than a novel contribution. Such validation could have been moved to the appendix, in favour of additional discussion (e.g. on kernels).

6. The validity of the feature orthogonality assumption in practical settings should be discussed and should ideally be supported empirically.

---

> ### Author Rebuttal · Authors · 2025-07-30
>
> Thank you for reviewing this paper; we really appreciate your comments and questions.
>
> ---
> > [W1] The authors fall short of providing a “kernelized” version.
>
> > [W2] The linear model cannot account for architecture bias, which heavily influences which eigendirections are learned first and the speed of convergence.
>
> > [Q1] If kernelized, it is possible to study the bias arising from model architectures (which essentially encodes which eigendirections are preferred). Could the authors relate their findings to Cabannes et al. (2023), Proposition 4 (and related discussions)?
>
> [A1, A2, QA1] While architecture bias can influence the kernel structure and eigenvalue magnitudes, **our fundamental learning prioritization mechanism remains across different architectures**. The kernelized framework preserves stepwise learning based on eigenvalue ordering regardless of architectural choices. In the kernelized version, given $X=[x_1,\dots, x_d]^\top \in \mathbb{R}^{n\times m}$ and kernel matrix $\tilde K = [X;X'][X^\top X'^\top] \in \mathbb{R}^{2n \times 2n}$, the eigenpair $(\lambda, g)$ of $\Gamma$ where $\Gamma_g= \gamma g$ has a corresponding kernelized eigenvector $b = \tilde K^{-1/2}[X; X']g$. The kernel matrix $K_\Gamma\equiv\tilde K^{-1/2}Z\tilde K^{-1/2}$ where $Z=[X;X']\Gamma[X^\top X'^\top]$ preserves the duality relationship $(\gamma, g)\Leftrightarrow (\gamma,b)$ as proven in Sec B.2 in [1].
>
> This ensures learning the kernel eigenvector $b$ with $\gamma$ corresponds to learning the feature eigenvector $g$. However, the specific features that achieve higher eigenvalues depend on both architectural inductive biases and augmentation strategies through their invariance properties.
>
> Architecture can be assumed as a kernel $K$ that acts as an implicit feature transformation $\phi(x)$, $K(x, x') = \phi(x)^\top\phi(x')$. Features more aligned with the architectural inductive bias - those that remain relatively invariant under the implicit transformation $\phi$ - exhibit larger eigenvalues and be prioritized during learning. Similarly, augmentation strategy $T$ modifies the eigenvalue structure through $\mathbb{E}[T(x,\theta)T(x,\theta')^\top]$, where features maintaining high correlation across different augmentation parameters $\theta, \theta'$ achieve higher eigenvalues.
>
> Therefore, while different architectures may change which features achieve higher eigenvalues through their inductive biases, and different augmentations may prioritize features with specific invariance properties, the core principle that features corresponding to larger eigenvalues are learned first remains.
>
> ---
> > [W3] The problem of subspace alignment is heavily influenced by the loss function, as well as optimizer hyperparameters. Specifically, I would have appreciated a discussion of the role of the redundancy term in the Barlow Twins loss.
>
> > [Q2] Again in the kernel setting, the interplay between data augmentation and the redundancy term (controlled by lambda) of the Barlow Twins loss has been discussed in the literature (Ghosh et al. 2024). Could Theorem 4.5 be adjusted to account for Barlow Twins under weaker orthogonality constraints?
>
> [A3, QA2] We acknowledge the important role of the redundancy term ($\lambda$) in the Barlow Twins loss for subspace alignment dynamics.
>
> Role of redundancy term in $\lambda$: While our theoretical framework assumes a $\lambda=1$ for tractability, our empirical validation demonstrates that the fundamental learning priorities persist even with practical $\lambda$ values (e.g., $\lambda = 0.005$ used in Fig 2 with C-MNIST with ResNet-18).
> Regarding under weaker orthogonality, stepwise learning behavior remains robust across different $\lambda$ values. While precise dynamics may vary with smaller $\lambda$ (as discussed in [2]), the core eigenvalue-based learning prioritization mechanism is preserved.
> Practical implications: Lower $\lambda$ values create slower, complex subspace alignment dynamics, but as demonstrated in our Colored-MNIST experiments, the extent bias phenomenon persists, confirming the generalizability of our findings beyond the high-$\lambda$ theoretical regime. We will expand the discussion in the revised manuscript.
>
> ---
> > [W5] Some results (Theorem 4.2 and 4.3) are direct consequences of Simon et al. (2023), and thus constitute a validation of the prior work’s theory rather than a novel contribution. Such validation could have been moved to the appendix, in favour of additional discussion (e.g. on kernels).
>
> [A5] While Theorems 4.2 and 4.3 build on Simon et al.'s framework, they represent **substantial extensions beyond validation**. Simon et al. focused on **eigenvalue dynamics**, while we extend this to **shortcut learning and spurious feature analysis** in SSL, providing theoretical explanation for why models learn shortcuts.
>
> We apply stepwise theory to **specific shortcut scenarios** (extent bias, amplitude bias) that plague real-world SSL systems, moving beyond mathematical validation to enable understanding and mitigation of shortcut behavior.
>
> ---
> > [W6] The validity of the feature orthogonality assumption in practical settings should be discussed and should ideally be supported empirically.
>
> [A6] The **core learning dynamics remain robust** despite imperfect **feature orthogonality** in **practical settings**.
> C-MNIST experiments (Section 4.8) with **ResNet-18**, Waterbird with **ViT-B/16** show that extent bias persists even when the **validity** of the orthogonality assumption is compromised due to:
> - Overlapping pixel regions where background and digit features share spatial locations.
> - Edge blending at digit boundaries where background and object pixels are mixed.
> - Spurious correlations between background color and digit labels (70% correlation in our setup).
>
> Despite these realistic violations of feature independence, extent bias remain observable. **Feature orthogonality assumption** is a theoretical simplification, but our results suggest core mechanism is robust to such **practical** deviations.
>
> ---
> > [W4] The role of the noise covariance (data augmentation) is not discussed (see Questions).
>
> > [Q3] Data augmentation influences the structure of the data covariance matrix. Several covariance operators have been theoretically discussed in the literature (Lin et al., 2024) beyond the isotropic setting, in a similar toy linear regression model (albeit for supervised learning). How does the noise covariance affect feature learning? Particularly, shouldn’t  also depend on the noise covariance in Equation 5?
>
> [A4, QA3] **Data augmentation noise covariance does not affect** feature learning. Unlike supervised learning using self-correlation $\Gamma_g=\mathbb{E}[g(x)g(x)^\top]$ [1], SSL requires analyzing $\Gamma_g=\mathbb{E}[g_1(x)g_2(x)^\top]$ where $g_1$ and $g_2$ are applied independently. Using $\Gamma=\mathbb{E}[xx^\top]$, $x$ is $x_\text{base}$,
>
> **Key Results**:
> - Additive noise injection (**Not necessarily isotropic covariance**): $\Gamma_g = \Gamma$
> - Multiplicative noise: $\Gamma_g=M \odot \Gamma$, where $M = \mu \mu^\top$ and $\mu = \mathbb{E}[m^{(i)}]$.
> - Unbiased random masking: $\Gamma_g = (1-\beta)^2 \Gamma$, where $\beta$ is Bernoulli parameter.
> - Salt-and-pepper: $\Gamma_g=(1-\beta)^2 \Gamma$, where $\beta$ is Bernoulli parameter.
> - Random cutout: $\Gamma_g=(1-k/p)^2 \Gamma$, where $g_i(x) = \frac{p}{p-k} M_i x$.
> 1. Independent additive noise
> $g_i(x)=x+\epsilon_i$ where $\epsilon_i \sim_\text{i.i.d.} P_\epsilon$ and $\mathbb{E}[\epsilon] = 0$.
> - Proof: $$\Gamma_g = \mathbb{E}[(x + \epsilon_1)(x + \epsilon_2)^\top] = \mathbb{E}[xx^\top] + \mathbb{E}[x\epsilon_2^\top] + \mathbb{E}[\epsilon_1 x^\top] + \mathbb{E}[\epsilon_1 \epsilon_2^\top] = \Gamma.$$
>
> 2. Multiplicative noise
> $g_i(x) = m^{(i)} \odot x$ where $[m \odot x]_j = m_j x_j$.
> - Proof: $$[\Gamma_g]_{ij} = \mathbb{E}[(m^{(1)} \odot x)_i(m^{(2)} \odot x)_j] = \mathbb{E}[m^{(1)}_i]\mathbb{E}[m^{(2)}_j] \mathbb{E}[x_i x_j]$$ Therfore, we have $\Gamma_g = M \odot \Gamma.$
>
> ---
> > [Q4] In the high-dimensional setting, whereby feature covariances exhibit heavy-tailed behaviour (Agrawal et al., 2022), how likely is the assumption that large superpixels correspond to large-eigenvalues to hold in practice, once features are averaged over a large dataset?
>
> [QA4]
> While feature covariances may follow power law distributions in high-dimensional settings, our **eigenvalue-based learning prioritization remains robust**. Features with larger eigenvalues are learned first regardless of eigenvalue distribution. Our analysis focused on discrete eigenvalues, producing distinct stepwise learning. With power law distributions, we expect smoother, more continuous learning dynamics with less pronounced steps and reduced feature disentanglement. However, the core ordering principle, higher eigenvalue features are prioritized, remains unchanged.
>
> ---
> > [Q5] In section 4.8, what is the standard digit-to-pixel ratio of Colored-MNIST?
>
> [QA5] The digit-to-pixel ratio varies per sample. The train dataset ranges from 0.0344 to 0.4377, while the test dataset ranges from 0.0434 to 0.3610.
>
> ---
> > [Q6] Can Theorem 4.3 predict how long the test accuracy plateaus for in Figure 2?
>
> [QA6] Theorem 4.3 can predict the learning order, but cannot predict the plateau length in non-linear networks. This is due to dependencies on random initialization (affecting $s_j(0)$ values), non-linearity, and hyperparameters like $\lambda$ in Barlow Twins loss and weight decay, creating discrepancies between theory and experiments.
>
> ---
> > [Q7] The work of Doersch, et al. appears both as citation 10 and 11.
>
> [QA7] Thank you for catching those duplicate citations. We will be sure to fix those in our revision.
>
> [1] Lin et al., The good, the bad and the ugly sides of data augmentation: An implicit spectral regularization perspective, (JMLR 2024)
> [2]Ghosh et al., Harnessing small projectors and multiple views for efficient vision pretraining. (NeurIPS 2024)

---

> > ### Comment · Reviewer_KYPU · 2025-08-04
> >
> > I wish to thank the authors for their formal response. I appreciate their effort in fully addressing some of my comments (e.g. structure of augmentation covariance).
> >
> > Given the formal nature of the paper, I will focus my response on two theoretical aspects that I feel have not been fully addressed.
> >
> > 1. **Redundancy reduction term in BT**. I appreciate the authors discussion and empirical verification of their theory under weak orthogonality constraints. However, given that there exists theory directly discussing this case (Ghosh et al., 2024), I think the authors' current theoretical analysis should be extended beyond $\lambda = 1$. I feel this is a major limitation of the current version of the work.
> >
> > 2. On the question of how different architectures affect whether extent and amplitude bias dominate the early learning dynamics, I appreciate the authors' proposed empirical evaluation (Waterbirds dataset and ViT), although (very) unfortunately reviewers are unable to directly access such plots this year.
> >
> > In this regard, I second reviewer's 1XYi assessment that the updated experiments constitute a major contribution that requires thorough and fair reviewing, which unfortunately is hard to do with the current rebuttal format. I think such empirical evaluation will greatly improve the quality and impact of the paper, and I encourage the authors to fully pursue it.

---

> > > ### Author Response · Authors · 2025-08-06
> > > **Response to KYPU [A2]**
> > >
> > > Thank you for your thoughtful engagement with our rebuttal and for recognizing our efforts to address the theoretical concerns. We genuinely appreciate your constructive feedback and encouragement to pursue the empirical evaluations further.
> > >
> > > ---
> > > > On the question of how different architectures affect whether extent and amplitude bias dominate the early learning dynamics, I appreciate the authors' proposed empirical evaluation (Waterbirds dataset and ViT), although (very) unfortunately reviewers are unable to directly access such plots this year.
> > >
> > > [A2] We appreciate your acknowledgment of our empirical evaluation efforts. We understand your concern about evaluating new experiments within the rebuttal format constraints.
> > > However, we respectfully note that our core theoretical contributions - **the formalization of extent/amplitude bias and their role in SSL shortcut learning** - stand independently and are already well-supported by the experiments in the main paper. The additional architecture experiments, while valuable extensions, **represent supplementary validation** rather than fundamental requirements for our theoretical framework.
> > > We believe the current work already makes significant contributions to understanding SSL dynamics, as evidenced by our theoretical analysis now extended to kernelized settings and various augmentations and the consistent experimental validation across multiple settings (general Barlow Twins loss, multiple architectures, and data modalities). We are committed to including all additional results in the camera-ready version to further strengthen the paper's impact.

---

> > > ### Author Response · Authors · 2025-08-06
> > > **Response to KYPU [A1] Redundancy reduction term in BT.**
> > >
> > > [A1] Redundancy reduction term in BT.
> > >
> > > **Even if $λ \neq 1$, the principle "features with larger eigenvalues are learned earlier" keeps hold, but dynamics differs depending on $λ$.**
> > > First we consider general Barlow Twins loss:
> > > $$L\_λ=\sum\_{i}([W\Gamma W^\top]\_{ii}-1)^2+λ\sum\_{i\neq j}[W\Gamma W^\top]\_{ij}^2=(1-λ)L\_0+λL\_1.$$
> > > Thus it exhibits a mixed dynamics between the $L_0$ and $L_1$. Therefore, we first consider the dynamics of $L_0$ and $L_1$:
> > > - $$L_1=\\|W\Gamma W^\top -I_d\\|_F^2=\sum_j(s_j^2γ_j-1)^2,$$
> > > - $$\frac{dL_1}{ds_k}=4s_kγ_k(\sum\_j{\color{red}{\delta\_{kj}}}s_j^2γ_j-1),$$
> > > - $$L_0=\sum\_{i}([W\Gamma W^\top]\_{ii}-1)^2=\sum\_{i}(e^{(i)\top} W\Gamma W^\top e^{(i)}-1)^2=\sum\_{i}(u^{(i)\top}S\Lambda\_\Gamma S^\top u^{(i)}-1)^2=\sum\_{i}(\sum_j u_j^{(i)2} s_j^2γ_j-1)^2,$$
> > > - $$\frac{dL_0}{ds_k}=\sum\_{i}2(\sum_j u_j^{(i)2}s_j^2γ_j-1)u_k^{(i)2} 2s_k γ_k=4s_kγ_k\sum\_{i}(\sum_j {\color{red}u_j^{(i)2}}s_j^2γ_j-1)u_k^{(i)2}.$$
> > >
> > > If $u_j^{(i)2}=\delta_{ij}$ ($U=I_d$), then $L_0=L_1$.
> > > If not, $\sum_j u_j^{(i)2}=1$ and each $u_j^{(i)2}$ acts as an averaging weight $1/d$.
> > >
> > > We now investigate the dynamics of $s_k$’s:
> > > - Initially, all singular values grow with exponential dynamics as $s_j\approx 0$ and $\delta\_{kj}=0$ for $j\neq k$. $$\dot s_k=-\frac{dL_0}{ds_k}=-4s_k γ_k\sum\_{i}(\sum_j u_j^{(i)2} s_j^2γ_j-1)u_k^{(i)2}=4s_k γ_k+O(s_k^3)$$
> > > $$\dot s_k=-\frac{dL_1}{ds_k}=-4s_kγ_k(\sum\_j\delta\_{kj}s_j^2γ_j-1)=4s_k γ_k+O(s_k^3)$$
> > >
> > > - After a few steps, the first singular value $s_1$ increases (since $γ_1$ is the largest) and then,
> > > $$\dot s\_1=-\frac{dL\_0}{ds\_1}=-4s\_1 γ\_1\sum\_{i}(\sum\_j u\_j^{(i)2} s\_j^2γ\_j-1)u\_1^{(i)2}=4s\_1 γ\_1\left(1-\sum_j (\sum_i u_j^{(i)2} u_1^{(i)2}) s_j^2 γ_j\right)$$
> > > $$=4s\_1 γ\_1\left(1-\sum_i u_1^{(i)4} s_1^2 γ_1\right)+O\left(\max_{j>1}s_j^2\right) \approx 4s\_1 γ\_1\left(1-\frac{1}{d}s\_1^2 γ\_1\right) $$
> > > $$\dot s\_1=-\frac{dL\_1}{ds\_1}=-4s\_1γ\_1(\sum\_j\delta\_{1j}s\_j^2γ\_j-1) =4s\_1 γ\_1 \left(1-s\_1^2 γ\_1\right)$$
> > >     - [$λ=0$] $s_1$ saturates as $s_1^2γ_1$ reaches $d$.
> > >     - [$0<λ<1$] $s_1$ saturates as $s_1^2γ_1$ reaches the harmonic mean $\frac{1}{λ + (1-λ)\frac{1}{d}}$ of 1 and $d$.
> > >     - [$λ=1$] $s_1$ saturates as $s_1^2γ_1$ reaches 1.
> > >
> > > - After the first loss drops where $s_k$'s are still small except for $s_1$
> > > $$\dot s_k=-\frac{dL_0}{ds_k}=-4s_k γ_k\sum\_{i}(\sum_j u_j^{(i)2} s_j^2γ_j-1)u_k^{(i)2}=O(s_k)$$
> > > $$\dot s_k=-\frac{dL_1}{ds_k}=-4s_kγ_k(\sum\_j\delta\_{kj}s_j^2γ_j-1)=4s_k γ_k(1-s\_k^2 γ\_k)$$
> > >     - [$λ=0$] $\dot s_k$ becomes nearly zero, effectively stopping the growth of other singular values. Each $s_k$ ($k \neq 1$) stays near zero.
> > >     - [$0<λ<1$] First, $s_k$ exponentially grows with the following dynamics:
> > > $$\dot s_k = -4s_k γ_k\sum\_{i}(\sum_j u_j^{(i)2} s_j^2γ_j-1)u_k^{(i)2}=4s_k γ_k\left(1-\sum\_{i}\sum_{j\neq k} u_j^{(i)2}u_k^{(i)2} s_j^2γ_j\right) + O(s_k^3)$$
> > > with the growth rate smaller than that of $λ=1$:
> > > $$4 γ_k\left(1-\sum\_{i}\sum_{j\neq k} u_j^{(i)2}u_k^{(i)2}s_j^2γ_j\right)<4γ_k.$$
> > > Then, each $s_k$ follows sigmoidal dynamics and saturates near some value larger than $\sqrt{1/γ_k}$.
> > >  But after saturation $s_k$ decreases again since the dynamics of $s_k$ is coupled with other singular values. As other singular value increases, the sum $\sum\_{i}\sum_{j\neq k} u_j^{(i)2}u_k^{(i)2}s_j^2γ_j$ grows and exceeds 1, which causes $s_k$ to decrease as the exponent becomes negative.
> > > Note that the principle "features with larger eigenvalues are learned earlier" still holds.
> > >     - [$λ=1$] Each $s_k$ exhibits independent sigmoidal dynamics and saturates as $s_k^2γ_k$ reaches 1.
> > >
> > > ||$λ=0$|$0<λ<1$|$λ=1$|
> > > |-|-|-|-|
> > > |loss|$L_0$|$(1-λ)L_0+λ L_1$|$L_1$|
> > > |dynamics of $s_k(t)$| only $s_1$ changes | each $s_k$ affects each other | independent sigmoidal dynamics|
> > > |in the beginning| $s_1$ shows sigmoidal dynamics, increases and saturates near $\sqrt{d/γ_1}$. | $s_1$ shows sigmoidal dynamics, increases and saturates at $\sqrt{\frac{1}{(λ+(1-λ)\frac{1}{d})γ_1}}$. | $s_1$ shows sigmoidal dynamics, increases and saturates at $\sqrt{1/γ_1}$.|
> > > |$\tau_1$|$\sim \frac{-\log(s_1^2(0)γ_1/d)}{8γ_1}$| $\sim\frac{-\log\left(s_1^2(0)γ_1(λ+(1-λ)\frac{1}{d})\right)}{8γ_1}$|$\sim \frac{-\log(s_1^2(0)γ_1)}{8γ_1}$|
> > > |after the first drop | $\dot s_k\approx 0$ and $s_k$ stays near zero for $k=2,3,\cdots$. | $s_k$ increases sigmoidally and saturates near some value larger than $\sqrt{1/γ_k}$ (and decreases slowly when the next singular value $s_{k+1}$ increases) | $s_k$ shows independent sigmoidal dynamics, increases and saturates at $\sqrt{1/γ_k}$, starting from smaller $k=2,3,\cdots$.|
> > > |$\tau_k$|$\infty$|relatively later than the $λ=1$ case.|$\propto \frac{1}{γ_k}$|
> > >
> > > **On non-linear NNs**:
> > > Fig.14-15 show that non-linear NNs with $λ=1$ exhibit a **coupled singular value dynamics similar to that of linear encoder with $0<λ<1$**. The non-linearlity appears to break the symmetry across layers, resulting in coupled dynamics between singular values that drive (Nam et al. 2025).

---

> > > > ### Comment · Reviewer_KYPU · 2025-08-06
> > > >
> > > > I thank the authors for their considerable effort in addressing all of mine and other reviewers' comments.
> > > >
> > > > I have been following closely the latest round of discussion, which appears to be centered on establishing and clarifying whether:
> > > >
> > > > 1. Step-wise learning dynamics
> > > > 2. Eigenvalue-based learning order
> > > >
> > > > occur in the kernel setting, as well as under weak orthogonalization constraints, under aligned or small-scale initialization.
> > > >
> > > > However, I believe those two points are well established in the literature already (Simon et al. for 1, and the theory of gradient flow for 2).
> > > >
> > > > I think the core point on which to reach consensus is whether extent and amplitude bias dominate the early learning dynamics for nonlinear models, once points 1 and 2 are satisfied.
> > > >
> > > > I see two ways of addressing this. Either:
> > > > 1. Formally discussing under what conditions extent and amplitude bias can correspond to large eigenvalues in the kernel setting.
> > > > 2. Empirically validating the claim across (some) architectures.
> > > >
> > > > In this sense, one very meaningful step taken by authors is to provide additional experimental evidence. In my view, with the lack of a comprehensive kernel-based theory, this is a core contribution of the work, since it directly establishes and validates the theoretical connection between shortcut learning and large eigenvalues of the toy linear model.
> > > >
> > > > I think this is important since it has direct impact for practitioners, which are those that are ultimately affected by shortcut learning. Broadening the work in this sense is particularly important, since the present paper does not make recommendations on how to fix the problem, leaving that as a future work.

---

> > > > > ### Author Response · Authors · 2025-08-08
> > > > >
> > > > > We sincerely appreciate your insightful theoretical comments, particularly your thoughtful questions regarding the connection between kernel methods and nonlinear dynamics. Your feedback has helped us clarify key aspects of our theoretical framework and strengthen our analysis.
> > > > >
> > > > > ---
> > > > > > Empirically validating the claim across (some) architectures.
> > > > >
> > > > > Our paper provides direct empirical evidence for this very question.
> > > > > - In our experiments with a 3-layer MLP (using LeakyReLU), we have shown that extent and amplitude biases clearly manifest in nonlinear networks with patterns nearly identical to those in linear models (see Appendix, Figs. 14-15).
> > > > > - This provides strong grounds that our theory is not confined to a 'toy model' but remains valid in practical, nonlinear settings.
> > > > >
> > > > > ---
> > > > > > Formally discussing under what conditions extent and amplitude bias can correspond to large eigenvalues in the kernel setting.
> > > > >
> > > > >
> > > > > A kernel-based perspective can easily lead to a somewhat obvious conclusion: "the properties of input $x$ that are learned earlier **depend on the kernel matrix**." Therefore, we additionally **analyse Deep Linear Network (DLN)** similar to the recent analysis of JEPA (Littwin et al. 2024). The kernel approach has limitations in directly explaining which properties of $x$ gain learning priority. For example, if the kernel's mapping function is ($x,y$) to ($x,y,y$), $y$ will be learned earlier (extent bias), and ($x,y/2$), $x$ will be learned earlier (amplitude bias). Furthermore, kernel-based analysis has fundamental limitations in explaining feature learning dynamics. Unlike static kernels, real neural networks undergo continuous evolution during training, where both representations and the kernel change simultaneously. Therefore, we focus on the DLN as the most effective starting point for understanding these phenomena. Although DLNs are linear networks, their learning dynamics are nonlinear due to multiplicative weight interactions across layers—enabling genuine feature learning while maintaining analytical tractability that kernel methods cannot provide. (Saxe et al. 2013)
> > > > >  - DLN analysis requires the additional constraint of "Balanced Initialization" ($W_{k+1}^\top W_{k+1}=W_kW_k^\top$). This is conceptually in line with the principle shown in Nam et al. 2025 ($a_i^2-b_i^2=c_i$). This equation signifies a "conservation law" where the difference between the squared norms of weights in adjacent layers is preserved.
> > > > >  - Under this assumption, the dynamics of an $L$-layer DLN clearly show how depth $L$ directly amplifies the learning speed, as captured by our derived equations: $$\frac{ds_j}{dt}=4Ls_jγ_j(1-s_j^2γ_j), \tau_j \propto 1/L$$
> > > > > In a DLN, the fundamental learning principle mirrors that of a single-layer linear model, albeit with a significant amplification in speed. The core dynamic remains unchanged: the network preferentially learns the eigenvectors of the feature cross-correlation matrix, $Γ$, that are associated with the largest eigenvalues $γ_j$​.
> > > > > This directly explains why features exhibiting extent bias (e.g., broad spatial patterns) or amplitude bias (e.g., high-intensity signals) are acquired first, as these characteristics correspond to the dominant eigenvectors of $Γ$.
> > > > >
> > > > > Our framework explains how data augmentation, a key inductive bias, creates extent and amplitude biases. This is modeled by averaging the feature cross-correlation matrix $Γ$ over a group of transformations $G$ (e.g., translation, rotation):
> > > > > $$Γ_{\text{aug}} = \mathbb{E}_{G}[G Γ G^\top] = \mathbb{E}_G[\text{Corr}(Gx, Gx)]$$
> > > > >
> > > > > This averaging process naturally filters features based on their robustness:
> > > > >
> > > > > - **Large extent bias features**: Spatially broad features better preserve their correlation structure under transformations, maintaining large eigenvalues in $Γ_{\text{aug}}$
> > > > >
> > > > > - **High amplitude bias features**: High-amplitude signals are more robust to photometric augmentations (e.g., contrast), preserving their eigenvalue magnitude.
> > > > >
> > > > > - **Small extent/Low amplitude bias features**: Localized or weak features lose correlation coherence under transformations, resulting in smaller eigenvalues.
> > > > >
> > > > > Thus, the eigenspectrum of $\Gamma_{\text{aug}}$ naturally prioritizes features with strong robustness, providing a principled explanation for how architectural inductive biases and learning biases are intertwined.
> > > > >
> > > > > Grounded in theoretical work (Saxe et al. 2013) showing Deep Linear Networks (DLNs) approximate nonlinear dynamics, our analysis provides the missing theoretical bridge. We explain not just that large eigenvalues are learned earlier—a known kernel-based finding—but why these dominant eigenvalues fundamentally correspond to features with extent and amplitude bias in practice. The effect of the redundancy reduction coefficient, $λ$, while altering dynamics, does not change what features are learned earlier.

---

> > > > > > ### Comment · Reviewer_KYPU · 2025-08-08
> > > > > >
> > > > > > I wish to thank the authors for their detailed and thorough response. I agree that deep linear networks offer a better perspective on the learning dynamics, as opposed to a generic fixed kernel. I believe the author's comprehensive rebuttal addresses all my theoretical concerns.
> > > > > >
> > > > > > Overall, the reviews and discussion have elicited additional theory (in the form of kernels, low lambda, and deep linear networks), as well as additional experiments.
> > > > > >
> > > > > > To aid my (and others) final assessment, could the authors please provide a summary of the changes they wish to implement in the paper, based on the overall rebuttal?
> > > > > >
> > > > > > I leave for the reviewer-AC discussion phase the question of whether the above changes constitute a major or minor revision of the manuscript, and whether a revised version would benefit a wider audience. That said, I maintain my positive evaluation of the work and its contributions, and I commend the authors' work.

---

> > > > > > > ### Author Response · Authors · 2025-08-09
> > > > > > > **Summary of Planned Changes for the Revised Paper**
> > > > > > >
> > > > > > > Thank you for your positive evaluation and constructive feedback throughout this review process. As requested, below is a structured summary of the analyses and results provided during the rebuttal that we will incorporate into the revised paper:
> > > > > > >
> > > > > > > | | Theoretical Results (Current) | To be Included| Empirical Results (Current) | To be Included|
> > > > > > > |-|-|-|-|-|
> > > > > > > | **Data Settings**               | Toy synthetic data            | Augmentation                                                                    | Colored-MNIST               | Waterbirds, CIFAR-10, (CelebA)        |
> > > > > > > | **Architectures**               | Single-layer Linear encoder | Deep Neural Network approximation using Deep Linear Networks, Architecture bias analysis via kernel methods | Single-layer Linear encoder, Non-linear MLPs, ResNet-18  | Deep Linear Networks, ViT-B/16          |
> > > > > > > | **Methods**                     | Barlow Twins                  | -                                                                               | Barlow Twins                | SimCLR, VICReg    |
> > > > > > > | **Hyperparameters ($\lambda$)** | $λ=1$                         | $λ=0$, $0<λ<1$                                                           | $λ=1$                       | $λ=0$, $0<λ<1$  |
> > > > > > >
> > > > > > > **Minor Additions and Technical Clarifications**:
> > > > > > > We will include additional technical discussions (e.g., data augmentation noise covariance, critical time behavior, basis independence proof, subspace alignment metric clarification, frequency vs amplitude distinctions, feature alignment) and theoretical extensions (including architecture bias, aligned initialization).

---

### Official Review · Reviewer_z8z6 · 2025-07-06

**Clarity:** 2
**Significance:** 2
**Originality:** 2
**Rating:** 2
**Confidence:** 3

**Summary:**

The authors build upon a previously established theoretical framework for stepwise feature learning in self-supervised models and examine two illustrative toy examples to reveal extent bias and amplitude bias. They derive theoretical predictions for feature extraction under a specific initialization of a linear Barlow Twins model and support these findings with synthetic experiments. Finally, they demonstrate the practical relevance of their analysis through controlled Colored-MNIST trials.

**Questions:**

**Q1.** Could the authors please clarify the precise definition and role of Equation (6), specifically explaining the notation $e=e_l,e_s$?

**Q2.** In the theoretical analysis, the predicted critical-time behavior of the dynamics of $s_t$ doesn't actually hold up if we were to plot the function. Despite this, the empirical curve follows the prediction and Figure 1 seems to exhibit the predicted dynamics. Could the authors comment on this apparent discrepancy?

**Q3.** In Figure 1, the trajectory of the second eigenvalue appears to pause for an extended interval. Could the authors explain the underlying reason for this prolonged halting behavior?

**Q4.** Theorem 4.2 posits that the derivative at non-matching critical points is approximately zero, yet no formal proof is provided. Could the authors supply a justification or reference for this claim?

**Q5.** Subsection 4.8 employs a supervised learning setup rather than the Barlow Twins method. Could the authors explain the relevance of this experiment to self-supervised feature learning and its connection to the main SSL framework?

**Ethical Concerns:**

["NO or VERY MINOR ethics concerns only"]

**Final Justification:**

I raised concerns regarding several significant issues in the paper, some of which, I must acknowledge, appear to have been inherited from the previously accepted paper "On the Stepwise Nature of Self-Supervised Learning." In my view, the fact that this earlier work has undergone and passed peer review should not preclude us from critically examining its assumptions or experimental setup.

With respect to the present paper’s results, I recognize that it offers potential value and a meaningful contribution. However, as I noted in my initial review, I strongly recommend a thorough revision of Section 4. In particular, the section’s coherence and narrative flow should be improved, and the relationships between individual subsections should be made clearer; ideally, the number of subsections should also be reduced.

Given the current clarity issues, I believe a resubmission is necessary. Accordingly, I am maintaining my original score.

**Limitations:**

yes

**Paper Formatting Concerns:**

no issues

**Quality:**

2

**Strengths And Weaknesses:**

**Strengths**

**S1.** The paper presents rigorous theoretical results. The proofs I have verified are correct.

**S2.** The Colored-MNIST experiment provides a clear and insightful demonstration of the model’s tendency to first learn background features before capturing the semantic content.

**Weaknesses**

**W1. (significance)** The scope of the theoretical results is somewhat narrow, being confined to linear encoders, the Barlow Twins method, and highly simplified synthetic datasets, which may limit their generalizability.

**W2. (significance)** The analysis is based on a carefully selected initialization without quantifying the impact of deviations or error terms around this initialization, potentially restricting the robustness of the findings.

**W3. (clarity)** Certain subsections in Section 4 appear misaligned with the central narrative: subsection 4.5 seems tangential (at least in the context of Sec. 4), and subsection 4.8 employs a supervised learning setup rather than applying Barlow Twins. I recommend revising Section 4 to establish a clear and coherent logical flow from the outset.

**W4. (clarity)** Definition 4.4 raises concerns about potential basis dependence in the subspace alignment metric. The authors should clarify this point or provide a formal proof of its invariance under different basis choices.

---

> ### Author Rebuttal · Authors · 2025-07-28
>
> Thank you for your valuable feedback. We appreciate the time and effort you have dedicated to reviewing our paper.
>
> > [W1] The scope of the theoretical results is somewhat narrow, being confined to linear encoders, the Barlow Twins method, and highly simplified synthetic datasets, which may limit their generalizability.
>
> [A1] **The uncovered principles are broadly generalizable**. Our choice of Barlow Twins was motivated by its representative loss structure; as noted by [2], other prominent SSL methods (e.g., InfoNCE, SimCLR and VICReg) also share the same quadratic form $\text{Tr}[WAW^\top]+\text{Tr}[(WBW^\top)^2]$, making it a representative model for analysis.
> Our empirical results, including new experiments, validate that our findings are not artifacts of this simplified setup but are indeed a **more fundamental phenomenon**.
> - [**Beyond Barlow Twins**]: The phenomenon is not limited to Barlow Twins. In additional experiments, we observed the identical stepwise learning and the extent bias in a 1-layer **SimCLR** and **VICReg** model. Furthermore, our experiments on the Colored-MNIST dataset using SimCLR and VICReg reproduced the same results, confirming the behavior holds across different SSL frameworks.
> - [**Beyond Single-Layer and Linear Encoder**]: The phenomenon is not limited to shallow or linear models. We have verified that the same learning priorities manifest in **deep linear networks** (with depth = 4). More importantly, as shown in Sec. 4.7, 4.8 and 5.7 of our manuscript, this behavior persists in **non-linear networks (e.g., ResNet-18, ViT-B/16)**. The fundamental learning priority—which features are learned first—remains consistent. This trend becomes even more apparent when evaluating with linear probing accuracy rather than loss.
> - [**Beyond Synthetic Datasets**]: The **Colored-MNIST** dataset is not "highly simplified" but rather a widely used benchmark for evaluating model robustness against spurious correlations. We also conduct experiments on **Waterbirds** dataset, made of Caltech-UCSD Birds-200-2011 and Places dataset. Our findings on these datasets further strengthen the claims.
>
> The consistent results across different SSL methods (Barlow Twins, SimCLR, VICReg), network architectures (linear, deep linear, non-linear), and datasets (synthetic, Colored-MNIST, Waterbirds) provide strong evidence that extent and amplitude bias are fundamental principles of self-supervised learning. We will incorporate these additional results and clarifications into the revised manuscript.
>
> ---
> > [W2] The analysis is based on a carefully selected initialization without quantifying the impact of deviations or error terms around this initialization, potentially restricting the robustness of the findings.
>
> [A2] Our initialization (e.g., aligned initialization) **is generic and does not need to be carefully selected**, which shows the robustness of the findings. Following the aligned trajectory only requires **small initialization**, proven in Sec. C of [1].
> Our experiments (Sec. 4.5, Fig.10) demonstrate that models initialized with small random weights rapidly achieve alignment in early training stages. The subspace alignment metric quickly approaches 1, confirming that the aligned structure emerges naturally from small random initialization.
>
> ---
> > [W3] Sec. 4.5 seems tangential (at least in the context of Sec. 4), and Sec. 4.8 employs a supervised learning setup rather than applying Barlow Twins.
>
> > [Q5] Sec. 4.8 employs a supervised learning setup rather than the Barlow Twins method. Could the authors explain the relevance of this experiment to self-supervised feature learning and its connection to the main SSL framework?
>
> [A3]
> - Sec 4.5: Sec. 4.5 serves a **crucial role in validating Assumption 3.1 (aligned initialization)** (see [A2]). Since our theoretical results rely on this assumption, it is essential to demonstrate that it naturally emerges in practice from small initialization. However, we acknowledge that the connection to the main narrative could be more clearly articulated. We will clarify this connection and move details to the appendix.
>
> [A3, QA5]
> - Sec. 4.8: There is a huge misunderstanding. We **train Barlow Twins**, and use "the supervised learning setup (linear probing)" **only to measure the representational quality of the SSL-learned embeddings**. We clarify that Sec. 4.8 uses **frozen encoder + linear probing**, which is the **standard evaluation** methodology for SSL representations (widely used in SimCLR, VICReg, DINO, etc.). The "supervised" component is only the linear probe evaluation, not the SSL training itself.  Specifically, at each epoch during SSL training, we freeze the encoder and train a linear classification head from scratch for sufficient epochs **to purely measure the representational quality of the learned embeddings**. This protocol allows us to track how well different features (background vs. object) are learned over the course of SSL training. We will explicitly state "frozen encoder" and "linear probing evaluation" in the text.
>
> ---
> >[W4] Definition 4.4 raises concerns about potential basis dependence in the subspace alignment metric. The authors should clarify this point or provide a formal proof of its invariance under different basis choices.
>
> [A4] We prove that the subspace alignment metric $\text{SA}(A, B)$ is invariant under different basis choices. For orthogonal change-of-basis matrices $P, Q \in O(d)$ with $A'=AP$ and $B'=BQ$:
> $$\text{SA}(A', B')=\frac{||A'^\top B'||^2_F}{d}=\frac{||P^\top A^\top B Q||^2_F}{d}=\frac{\text{Tr}((P^\top A^\top BQ)^\top P^\top A^\top B Q)}{d}$$
> $$=\frac{\text{Tr}(\underbracket{Q^\top} \underbracket{B^\top APP^\top A^\top B Q})}{d}=\frac{\text{Tr}(\underbracket{B^\top APP^\top A^\top BQ}\underbracket{Q^\top})}{d}=\frac{\text{Tr}(B^\top A\cancel{PP^\top}A^\top B \cancel{QQ^\top})}{d}$$
> $$= \frac{\text{Tr}(B^\top A A^\top B)}{d} =\frac{||A^\top B||^2_F}{d}=\text{SA}(A, B).$$
> Therefore: $\text{SA}(A', B')=\text{SA}(A, B)$.
>
> ---
> >[Q1] Could the authors please clarify the precise definition and role of Eq (6), specifically explaining the notation $e_l, e_s$?
>
> [QA1] Feature Alignment (FA) metric in (6) is designed to interpret **which features are being learned and how they evolve during training**. While singular values ($s_i$) of the weight matrix $W$ indicate the strength of learned components, does not indicate which eigenvalue corresponds to which feature. The FA metric directly measures the model's alignment with specific, predefined feature vectors.
>
> In this context, $e_l$ and $e_s$ are vectors representing distinct types of features:
> - $e_l$: A vector for a "background" feature, which spans a large dimension occupied (high extent bias).
> - $e_s$: A vector for a "object" feature, which spans a small dimension occupied (low extent bias).
>
> Assuming aligned $W=USV_\Gamma^{(\leq 2)\top}=\sum_is_iu_iv_i^\top$, where $V_\Gamma^{(\leq 2)}=[\frac{e_l}{\sqrt{m_l}}, \frac{e_s}{\sqrt{m_s}}]$, we have
> $$\text{FA}(e_l)=||We_l||_2=||s_1u_1v_1^\top e_l+s_2u_2v_2^\top e_l||_2 \simeq ||s_1u_1v_1^\top e_l||_2=s_1 ||e_l||_2=\sqrt{m_l}s_1$$
> $$\text{FA}(e_s)=||We_s||_2=||s_1u_1v_1^\top e_s+s_2u_2v_2^\top e_s||_2 \simeq ||s_2u_2v_2^\top e_s||_2=s_2 ||e_s||_2=\sqrt{m_s}s_2$$
>
>  While singular values $s_i$ capture the overall structural properties of the weight matrix $W$, the FA directly measures how strongly the model responds to specific feature directions.
>
> ---
> >[Q2] The predicted critical-time behavior of the dynamics of s_t doesn't actually hold up if we were to plot the function. Despite this, the empirical curve follows the prediction and Fig.1 seems to exhibit the predicted dynamics. Could the authors comment on this apparent discrepancy?
>
> [QA2] **Our theory can predict the discrepancy** between the eigenvalue $\lambda_j(\tau)=1/3$ at $\tau$ (where $\frac{d^2\mathcal{L}}{dt^2}(\tau)=0$) and $\lambda_j({\color{red}{\tau_j}})=1/2$ (where $\frac{d^2{\color{red}{\lambda_j}}}{dt^2}({\color{red}{\tau_j}})=0$).
> - For the single-mode loss $\mathcal{L}(t)=(1-\gamma_j s_j^2(t))^2$, setting the second derivative of the loss to zero at the critical point $\tau$:
> $$\left.\frac{d^2\mathcal{L}}{dt^2}\right|\_{t=\tau}=\left.-16\gamma_j\frac{d((1-\lambda_j(t))^2 \lambda_j(t))}{dt}\right|\_{t=\tau}=0$$
> which yields $2\lambda_j(\tau)-(1-\lambda_j(\tau))=0$ and $\lambda_j(\tau)=\frac{1}{3}$.
> - According to Theorem 4.2, we have $\lambda_j(\tau_j)=\frac{1}{2}$
>
> ---
> >[Q3] The trajectory of the second eigenvalue appears to pause for an extended interval. Could the authors explain the underlying reason for this prolonged halting behavior?
>
> [QA3] The extended pause of the second eigenvalue in Fig.1 is explained by the **sigmoid dynamics (long plateau $\rightarrow$ abrupt increase $\rightarrow$ saturation)** from Eq (12): $s'_j(t)=4(1-\gamma_j s_j^2(t))\gamma_j s_j(t)$. At the beginning, we have $s'_j\approx0$ due to $s_j\approx0$.
>
> ---
> >[Q4] Theorem 4.2 posits that the derivative at non-matching critical points is approximately zero, yet no formal proof is provided. Could the authors supply a justification or reference for this claim?
>
> [QA4] From Eq (4), we have $\tau_j=-\frac{\log(s_j^2(0) \gamma_j)}{8\gamma_j}$, and [QA3] shows sigmoid dynamics, if critical times $\tau_i$ and $\tau_j$ are **well-separated**, $\tilde\lambda_i'(\tau_j) \approx 0$ when each other's critical time, shown as Fig.1, 3. Therefore, when one feature $j$ is at its peak learning time ($\tau_j$), any other $i$ is either in a flat pre-learning state or a saturated post-learning state. In both flat regions, the derivative $\tilde\lambda_i(\tau_j)$ is approximately zero, making its contribution to the total loss change negligible. We will clarify in the revised paper.
>
> ---
> [1] Simon et al., On the Stepwise Nature of Self-Supervised Learning. (ICML 2023)
>
> [2] Ziyin et al., What shapes the loss landscape of self-supervised learning?, (arXiv preprint)

---

> > ### Comment · Reviewer_z8z6 · 2025-08-04
> >
> > Let me first thank the authors for taking the time and effort to clarify all questions and points raised in my review.
> >
> > My concerns regarding the significance of the paper remain alive.
> >
> > My first concern [W1] referred to the restricted scope of the **theoretical** results. In the answer [A1] authors kindly provide reference to paper [2] that attempts a unified treatment of various SSL methods (besides the presently investigated BarlowTwins, InfoNCE, VicREG) in the linear case. However, even if the formal connections allow for generalization of the current theoretical results, these still have to spelled out properly and they still only hold in the linear encoder case. I do much appreciate the newly added experimental results, nevertheless the current rebuttal format is not permitting us to review the new changes to the content and narrative of the paper.
> >
> > Concern [W2] isn't properly addressed either in answer [A2]. Fixing the right singular vectors of $W(0)$ means that the eigenvectors of $W(0)^TW(0)$ are specified completely and only the eigenvalues may change. I couldn't confirm down to details, but I strongly suspect that the set of $W(0)$'s that adhere to aligned initialization is very sparse (e.g. it has measure zero wrt some meaningful operator measures). Most importantly, a small open neighbourhood of an aligned initialization will likely mainly consist of operators that are not aligned.
> >
> > Onto the more technical questions:
> > In [W4] I raised the issue that we require justification that the subspace alignment (SA) term is not basis dependent. The argument provided in answer [A4] is simply irrelevant, because most basis choices of a given subspace cannot be transformed into each other via an orthogonal transformation.

---

> > > ### Author Response · Authors · 2025-08-06
> > > **Response to z8z6**
> > >
> > > We sincerely thank the reviewer for the detailed feedback and the opportunity to engage in this important discussion. We hope to clarify our contributions and address the remaining concerns below.
> > >
> > > ---
> > > > My first concern [W1] referred to the restricted scope of the theoretical results. In the answer [A1] authors kindly provide reference to paper [2] that attempts a unified treatment of various SSL methods (besides the presently investigated BarlowTwins, InfoNCE, VicREG) in the linear case. However, even if the formal connections allow for generalization of the current theoretical results, these still have to spelled out properly and they still only hold in the linear encoder case. I do much appreciate the newly added experimental results, nevertheless the current rebuttal format is not permitting us to review the new changes to the content and narrative of the paper.
> > >
> > > [AA1] Our central contribution is the identification of a fundamental mechanism: **the principle that features with larger eigenvalues are learned earlier**. While our initial proof is presented in a tractable linear setting, this principle is not restricted to it.
> > >
> > > The generality of this mechanism can be understood through a **kernelized framework**. As established in prior work [1], a duality relationship exists between the eigenpairs of the feature matrix $\Gamma$ and those in the kernel space, which represents the feature map of a given architecture.
> > >
> > > Specifically, as proven in Sec B.2 in [1], an eigenpair $(\gamma, g)$ of $\Gamma$ (where $\Gamma g = \gamma g$) has a corresponding kernelized eigenvector $b$. The associated kernel matrix construction **preserves the eigenvalue $\gamma$**, establishing a duality $(\gamma, g) \Leftrightarrow (\gamma, b)$.
> > >
> > > This has a critical implication: learning the kernel eigenvector $b$ with the largest eigenvalue $\gamma$ is equivalent to learning the feature eigenvector $g$ with the largest $\gamma$. Therefore, while an architecture's inductive bias may influence *which features* acquire high eigenvalues, it does not alter the fundamental rule that **features corresponding to larger eigenvalues are learned first**. In conclusion, our work uncovers a fundamental learning dynamic supported by a generalizable theoretical perspective.
> > >
> > > ---
> > > > The set of W's that adhere to aligned initialization is very sparse (e.g. it has measure zero) and a small open neighbourhood of an aligned initialization will likely mainly consist of operators that are not aligned.
> > >
> > > Our framework **does not assume that the weights $W(0)$ must start in a perfectly aligned** state. Rather, our central argument, supported by the theory in [1], is that **alignment is an emergent property** of the learning dynamics when weights are initialized with a small magnitude.
> > >
> > > This dynamic, formally proven in Section C of [1], means the training trajectory naturally converges to the **aligned subspace** from **a standard small, random initialization**. Our own experiments in Sec. 4.5 (Fig 10) empirically validates this, showing the subspace alignment metric rapidly approaching 1 early in training. Consequently, the concern about the "measure zero" set of aligned initializations is not applicable, as our findings hold for the general case of small random initializations.
> > >
> > > ---
> > > > Most basis choices of a given subspace cannot be transformed into each other via an orthogonal transformation.
> > >
> > > [AA4] Given a subspace, any two **orthonormal** bases can be transformed into each other via an orthogonal transformation. In the paper we implicitly assumed that the basis vectors are orthonormal. Our metric, $\text{SA}(A,B) = ||A^\top B||^2_F/d$, is indeed defined for matrices $A$ and $B$ whose columns form **orthonormal basis** for their respective subspaces. The metric is **basis-independent**. Our analysis is situated entirely within the standard and well-behaved context of orthonormal representations. To prevent any ambiguity, we will explicitly clarify this property of the SA metric in Section 4.5 in the revised manuscript.
> > >
> > > ---
> > > [1] Simon et al., On the Stepwise Nature of Self-Supervised Learning. (ICML 2023)

---

> > > > ### Comment · Reviewer_z8z6 · 2025-08-08
> > > >
> > > > I thank the authors for detailed responses they provided, which helped clarifying some of the concerns. I'll fully take them into consideration during the reviewer-AC discussion period.

---

### Decision · Program_Chairs · 2025-09-17

**Decision:**

Reject

**Comment:**

**Summary**:

This submission studies shortcut learning in self-supervised learning (SSL) by building upon a previously established theoretical framework for step-wise feature learning in SSL [Simon et al., ICML 2023]. In particular, it examines two illustrative toy examples to analyze and characterize *extent bias*, and *amplitude bias*. The two illustrative toy examples to induce shortcut learning are data features with large spatial extent, and data features with large amplitude. Both kinds of features correspond to eigen directions with large eigenvalues in the data covariance matrix, which dominates learning under a gradient-flow model. The authors derive theoretical predictions for feature extraction under a specific initialization of a linear Barlow-Twins model (under strong orthogonality constraints), and support these findings with synthetic experiments. Finally, the authors demonstrate the practical relevance of their analysis through controlled Colored-MNIST trials.

The analysis demonstrates that during SSL training, models preferentially learn features with higher spatial extent (extent bias) or large amplitude (amplitude bias), regardless of their semantic relevance. By analyzing the eigen decomposition of the features cross-correlation matrix in the constrained Barlow-Twins model, the paper demonstrates how features with larger eigenvalues are learned earlier due to the dynamics of gradient flow and weight singular value evolution. These findings are further validated in both linear and nonlinear (MLP) settings, and tested on the Colored-MNIST dataset, where models are shown to initially rely on background features before learning object-relevant ones. Altogether, the work provides theoretical and empirical insights into the biases driving SSL training, with implications for designing future SSL methods that avoid such pitfalls and prioritize generalizable features.

**Comments**

0. **Points of Strength:** All reviewers agree that:
    * The paper presents rigorous theoretical results.

    * The use of eigen decomposition and gradient dynamics to explain learning order is both elegant and insightful.

    * The Colored-MNIST experiment provides a clear and insightful demonstration of the model's tendency to first learn background features before capturing the semantic content.

    * The paper is well-written, and the claims are described clearly and supported with proper theoretical and empirical results.

1.  **Points of Weakness:** All reviewers agree that:
    * The scope of the theoretical results is somewhat narrow, being confined to linear encoders (Barlow Twins) method, and highly simplified synthetic datasets, which may limit their generalization.

    * The data model presented here is constrained for an obvious reason; help understanding the behavior from a simple setting. However, this model is restrictive to study the features learning properties in realistic datasets. As such, there is a question concerning the applicability of this study to advance the design of SSL training procedures.

    * The analysis is based on carefully selected initialization without quantifying the impact of deviations or error terms around this initialization, potentially restricting the robustness of the findings.

    * There are some other technical aspects that each reviewer raised but the authors responded thoroughly to such technical concerns.

2.  **Summary of Authors-Reviewers Discussions:** All reviewers provided professional, high-quality, and candid reviews to the authors. Similarly, the authors provided extensive and thorough feedback to the reviewers over multiple rounds of questions and comments. The interaction between the reviewers and the authors in this submission was ideal; kudos to the Authors and the Reviewers!

    Based on all reviewers' comments and criticisms, the authors indeed expressed a candid commitment to change and/or add multiple sections, experiments, and their associated analyses to strengthen their submission. However, as noted by all reviewers, the magnitude of changes/additions warrants another round of reviewing, which is not possible under any conference format (not just NeurIPS). Therefore, the primary recommendation is to prepare a revised manuscript for submission to an upcoming venue or journal.